# DUMMY CONSISTENT PATH METHOD FOR ROBUST ATTRIBUTION WITH IRRELEVANT FEATURES

## ABSTRACT

Attribution methods are widely used to interpret deep neural networks by identifying the input features that contribute to model predictions. Among these, path-based approaches, which accumulate gradients along a path from a baseline to an input, are often preferred for their theoretical foundations such as the Aumann-Shapley value. However, in networks with rectified activations, these methods frequently assign importance to irrelevant features—violating the Dummy Player axiom—due to the shattered gradients problem. Therefore, we propose the Dummy Consistent Attribution Path Method (DCAPM), a novel path-based approach designed to strictly enforce the exclusion of irrelevant feature interactions. Unlike standard approaches, our method dynamically constructs a path that nullifies spurious attributions arising from feature interactions with dummies to ensure zero attribution at every step, thereby ensuring theoretical consistency. To validate the practical impact of this theoretical adherence, we introduce a new metric that quantifies the alignment between dummy consistency and explanation faithfulness. We demonstrate that while existing methods systematically fail to satisfy this consistency, our approach yields significantly more robust and faithful explanations by adhering to the axiom.

## 1 INTRODUCTION

Interpreting the decision-making processes of deep neural networks (DNNs) is indispensable for ensuring safety and reliability, particularly in high-stakes domains such as medical diagnosis and autonomous driving where model errors can lead to catastrophic consequences. In this context, input attribution methods, which identify the contribution of individual input features to model predictions, have emerged as essential tools. These methods are leveraged for debugging and refining models by uncovering unintended model behaviors, such as identifying biases or spurious correlations (De Coninck et al., 2024; Bilodeau et al., 2024; Anders et al., 2022; Adebayo et al., 2020; Schramowski et al., 2020; Rieger et al., 2020; Ross et al., 2017), Furthermore, their importance has extended to analyzing the reasoning capabilities of modern large language models (Zhao et al., 2024).

To provide rigorous feature-level explanations without definitive ground truth, the community has turned to game-theoretic approaches, specifically the Shapley value (Shapley, 1953). The Shapley value offers a theoretically grounded framework for fairly distributing the model prediction output among input features. Its reliability stems from satisfying a set of desirable axioms, including Efficiency and Dummy Player—the latter guaranteeing that features with no functional contribution receive zero attribution. Path-based attribution methods (Sundararajan et al., 2017), such as Integrated Gradients (IG), aim to extend these theoretical properties to DNNs by accumulating gradients along a path from a baseline to the input. However, the practical application of these methods to rectified DNNs is often compromised, yielding noisy and counterintuitive attributions. While prior research has attempted to mitigate these issues by refining paths or baselines (Sturmfels et al., 2020; Kapishnikov et al., 2019; Xu et al., 2020), these efforts have not fully resolved the underlying instability.

In this paper, we argue that such approaches address the symptoms rather than the root cause. We identify the critical bottleneck as the interaction between path integration and the intrinsic piecewise-linear nature of rectified DNNs (Chu et al., 2018). The piecewise-linear networks create highly fragmented decision regions—manifesting as the shattered gradient phenomenon (Balduzzi et al., 2017; Montufar et al., 2014)—which causes path integrals to erroneously aggregate non-zero

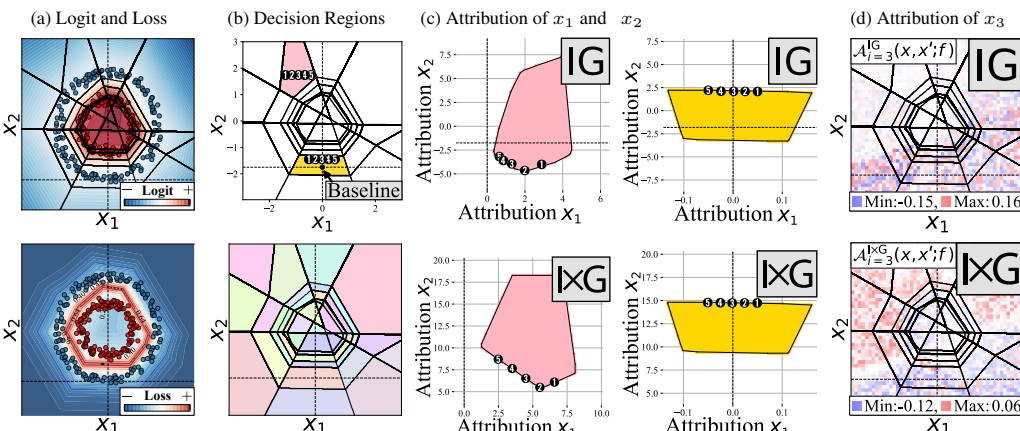

Figure 1: **Standard attribution methods violate fundamental axioms when a dummy feature is introduced.** We add a semantically irrelevant dummy feature $x_3$ to a simple 2D classification task to demonstrate the failure modes of existing methods. **(a) & (b) Model Behavior:** Logit/loss outputs and decision boundaries for the 2D signal features $x_1$ and $x_2$. Two linear regions are represented: the baseline-residing region (yellow) and the ordinary region (pink). Input points ❶ to ❺ represent a single-feature-axis-shift scenario. **(c) Violation of Dummy Consistency axiom:** Attributions for the true signal features, $x_1$ and $x_2$, are distorted by the presence of the dummy feature $x_3$. This violates the axiom that states attributions of signal features should remain consistent regardless of irrelevant inputs. **(d) Violation of Dummy Player Axiom:** The dummy feature $x_3$ receives non-zero attribution, incorrectly assigning importance to a random, irrelevant input. This violates the axiom that states the contribution of a dummy feature must be zero. Both path-based (IG, top) and non-path-based (I×G, bottom) methods fail. This experiment motivates our work by revealing a critical flaw in widely used attribution methods, thereby highlighting the need for a new approach that is robust to such dummy interactions. Additional experiments on TwoMoons dataset are presented in Appendix C.2, and the detailed experimental setup is in Appendix D.4.

attributions for functionally irrelevant features. Consequently, this leads to a direct violation of the Dummy Player Axiom.

Figure 1 illustrates a concrete failure case where existing methods fail to suppress the influence of dummy features. Ideally, features with no functional contribution should receive zero attribution; however, our analysis reveals that spurious dummy interactions cause these features to leak into the explanation. As shown in Figure 1 (d), the noise feature $x_3$ receives non-zero attribution, which clearly violates the Dummy Player Axiom. Furthermore, Figure 1 (c) indicates that the noise feature distorts the attribution of signal features ($x_1$ and $x_2$). By examining an input sequence shifted solely along individual feature dimensions (Figure 1 (b)), we observe concurrent changes in the attributions of $x_1$ and $x_2$. This suggests that the dummy feature $x_3$ induces counterintuitive behavior through spurious attribution arising from interactions with signal features. Leveraging the Shapley interaction index (Chang et al., 2025; Tsai et al., 2023; Janizek et al., 2021; Dhamdhere et al., 2020; Fujimoto et al., 2006; Grabisch & Roubens, 1999), we confirm that these interactions are inherently spurious when the path, a sequence of intermediate inputs evolving from the baseline to the original input (see Row 2 in Figure 2), traverses distinct piecewise linear regions. Such interactions violate the Dummy Consistency Axiom's requirement for fair contribution allocation. Detailed empirical validation of this phenomenon is provided in Appendix C.10.

To resolve this axiomatic violation, we propose the Dummy Consistent Attribution Path Method. This method dynamically constructs a path designed to nullify spurious dummy interactions while strictly adhering to the Dummy Player and Dummy Consistency axioms. Our approach employs a two-stage path generation process. First, we utilize a gradient-based mask optimization to identify and decouple dummy features from signal features. Second, we implement an explicit mechanism to nullify interactions between dummies and signals along the path. This procedure ensures that only relevant features contribute to the attribution, yielding explanations that are both reliable and axiomatically consistent.

Our main contributions are as follows: **(1)** We propose the *Dummy Consistent Attribution Path Method* (DCAPM), a path-finding approach that uniquely enforces the Dummy Player and Dummy Consistency axioms, thereby resolving the shattered gradient problem plaguing the rectified DNNs. **(2)** We introduce a novel metric designed to quantify the violation of the Dummy Consistency axiom in rectified DNNs. We demonstrate that, while this metric is orthogonal to existing faithfulness evaluations, reducing such axiomatic violations positively influences explanation faithfulness, validating the practical utility of theoretical consistency. **(3)** We empirically demonstrate that DCAPM outperforms prior methods by strictly adhering to the Dummy Consistency axiom, resulting in significantly more robust and trustworthy attributions.

## 2 PRELIMINARIES

### 2.1 AXIOMATIC ATTRIBUTION AND PATH METHODS

Path-based attribution methods draw their theoretical foundation from the Aumann-Shapley (AS) framework (Aumann & Shapley, 1974), which extends the axiomatic properties of the Shapley value (Shapley, 1953) to models with continuous inputs. The AS value is computed using a path integral, where the contribution of each feature is aggregated along a path from a baseline input to the original input. This approach is generalized by the family of path methods (Friedman, 2004), which was extended by methods like IG (Sundararajan et al., 2017) for application to DNNs.

**Definition 1** (Path Function (Friedman, 2004)). *A **path function** is a mapping $\gamma : [0, \hat{t}] \times \mathcal{X}(N) \times \mathcal{X}(N) \to \mathcal{X}(N)$ for a finite stopping step $\hat{t} < \infty$. For a time step $t \in [0, \hat{t}]$, a baseline $\mathbf{x}' \in \mathcal{X}(N)$, and an input $\mathbf{x} \in \mathcal{X}(N)$, the mapping function $\gamma(t; \mathbf{x}, \mathbf{x}')$ generates an intermediate input state. It satisfies continuous and feature-wise non-decreasing in t, with boundary conditions from a baseline $\gamma(0; \mathbf{x}, \mathbf{x}') = \mathbf{x}'$ to the input $\gamma(\hat{t}; \mathbf{x}, \mathbf{x}') = \mathbf{x}$,*

where $\mathcal{X}(N)$ denotes the space of all possible inputs with $N$ features, and $\mathbf{x}'$ is a baseline which typically represents an uninformative input to the function, satisfying $f(\mathbf{x}') \approx 0$. Given a path function, the attribution for each feature is calculated by integrating its infinitesimal contributions along that path. The path-based attribution for a function $f$ is defined as follows:

**Definition 2** (Path-Based Attribution Method (Friedman, 2004)). *Given a path $\gamma(t) := \gamma(t; \mathbf{x}, \mathbf{x}')$ and a baseline $\mathbf{x}'$, the attribution method $\mathcal{A}_i^\gamma(\mathbf{x}, \mathbf{x}'; f)$ defines the contribution of feature $i$ as the Riemann–Stieltjes integral:*

$$\mathcal{A}_i^\gamma(\mathbf{x}, \mathbf{x}'; f) = \int_0^{\hat{t}} \frac{\partial f(\gamma(t))}{\partial \gamma_i(t)} \frac{\partial \gamma_i(t)}{\partial t} \, dt. \tag{1}$$

The Aumann–Shapley value corresponds to a straight-line path $\gamma(t; \mathbf{x}, \mathbf{x}') = \mathbf{x}' + t(\mathbf{x} - \mathbf{x}')$ and uses assumptions of non-negativity and $\epsilon$-monotonicity of the function outputs along the path (Aumann & Shapley, 1974). Although the partial derivatives exist almost everywhere since rectifier-based DNNs are piecewise linear, they lack continuous second-order derivatives, so their gradient vector fields are non-conservative, making the integral path-dependent. Consequently, the choice of the path $\gamma$ is critical for rectified DNNs. Prior studies have explored how to construct effective paths (Zhang et al., 2024; Jeon et al., 2023; Yang et al., 2023; Zhuo & Ge, 2024; Akhtar & Jalwana, 2023; Kapishnikov et al., 2021).

### 2.2 DUMMY PLAYER AND DUMMY CONSISTENCY AXIOMS

Among the several axioms of the Shapley value, our work focuses on two axioms that are fundamental to ensuring principled and fair attribution allocation, especially in the presence of irrelevant or *dummy* features. The first is the Dummy Player Axiom, which provides a formal basis for identifying such irrelevant features.

**Definition 3** (Dummy & Null Player (Shapley, 1953)). *A feature $i$ is a **dummy player** in $(N, f)$ if its marginal contribution is constant and independent of other features: $f(S \cup \{i\}) - f(S) = f(\{i\}), \ \forall S \subseteq N \setminus \{i\}$. A feature $i$ is a **null player** in $(N, f)$ if it has no functional influence on the output, regardless of the presence of other features: $f(S \cup \{i\}) = f(S), \ \forall S \subseteq N \setminus \{i\}$.*

For a null feature $i$, the attribution is zero, satisfying $\mathcal{A}_i(\mathbf{x}, \mathbf{x}'; f) = 0$[1]. The second, and central axiom to our work, is Dummy Consistency (Friedman, 2004), which states that removing a dummy player should not alter the attribution assigned to the remaining signal features.

**Definition 4** (Dummy Consistency (Friedman, 2004)). *An attribution method $\mathcal{A}$ is **dummy consistent** if, for any dummy feature $i \in N$ defined as $\frac{\partial f(\mathbf{x})}{\partial x_i} := 0$, the attributions of all other features $j \neq i$ remain unchanged when $i$ is removed from the game. Formally: $\mathcal{A}_j(\mathbf{x}, \mathbf{x}'; f) = \mathcal{A}_j(\mathbf{x}_{-i}, \mathbf{x}'_{-i}; f^{R_i^x})$ $\forall j \neq i$, where $\mathbf{x}_{-i}$ denotes the input with feature $i$ removed, and $f^{R_i^x}$ is the reduced function over the remaining features $\mathbf{x}_{-i}$.*

Theoretically, the Dummy Consistency axiom should be satisfied by path-based methods automatically since the partial derivatives with respect to a dummy feature $i$ along any path are identically zero, making the reduced function trivial $f^{R_i^x}(\mathbf{x}_{-i}) = f([\mathbf{x}_{-i}, 0_i])$ and the axiom holds by definition. However, in rectified networks, the shattered gradient phenomenon disrupts this ideal behavior. Gradients for dummy features can become non-zero due to unstable region transitions, causing path methods to violate Dummy Consistency in practice.

**Shattered Gradients Problem.** The Shattered Gradients (Balduzzi et al., 2017; Montufar et al., 2014) is the phenomenon observed in DNNs where the spatial correlation between gradients at nearby data points decays exponentially with network depth. As a result, the gradient signal loses its local structure and increasingly resembles uncorrelated white noise. The shattering of gradients is a direct consequence of the composition of rectifier non-linearities across many layers, due to factors such as piecewise constant gradients and the exponential growth of activation boundaries.

### 2.3 SHAPLEY INTERACTION INDEX

To analyze how features contribute jointly, we leverage the Shapley Interaction (Fujimoto et al., 2006; Grabisch & Roubens, 1999), a principled framework for quantifying the effect of feature pairs. The classical Shapley interaction index is defined for discrete set functions. To apply this to continuous functions like DNNs, a principled approximation is required. We adopt an approximation based on the difference of first-order partial derivatives, as proposed by (see Chang et al., 2025, Equation 10). The interaction index $\mathcal{I}_{i,j}$ between features $i$ and $j$ for a given coalition $T \subseteq N \setminus \{i, j\}$ is defined as:

$$\mathcal{I}_{i,j}(f, T) \approx \frac{\partial f(T \cup \{j\})}{\partial x_i} - \frac{\partial f(T)}{\partial x_i}. \tag{2}$$

This first-order Taylor approximation is necessary because higher-order derivatives in rectified networks are zero almost everywhere, making the computation of higher-order interactions intractable.

### 3 DUMMY CONSISTENT PATH METHOD

While prior path-based attribution methods rely on desirable axioms, they often fail to satisfy the Dummy Player and Dummy Consistency axioms when applied to modern rectifier-based networks, resulting in unreliable and noisy attributions. We attribute these failures to two distinct mechanisms:

**Violation of the Dummy Player Axiom via Shattered Gradients** The foundational assumption that a dummy feature yields a zero partial derivative ($\partial f / \partial x_i = 0$) is rarely met in practice. This failure stems from the *shattered gradient* phenomenon (Balduzzi et al., 2017), where the piecewise-linear nature of the rectified networks causes the dummy features to exhibit a distribution of noisy, near-zero gradients instead of true zeros (as empirically verified in Appendix C.9). This directly leads to violations of the Dummy Player axiom, as standard path methods erroneously aggregate these noisy gradients, assigning non-zero importance to functionally irrelevant features.

**Violation of the Dummy Consistency Axiom via Spurious Interactions** From the perspective of Shapley interaction, we identify that the contributions of feature interactions are captured through traversal across the different linear regions and that the model gradients can easily assign non-zero

---

[1]Much of the XAI literature uses the term *dummy* to refer to the stricter null player case. Following this convention, our work addresses the null player condition under the terminology of the Dummy Player axiom.

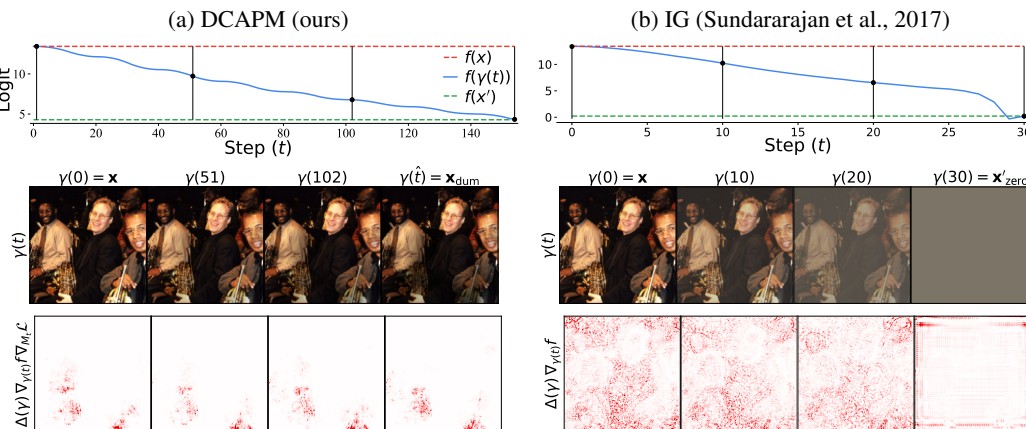

Figure 2: **Comparison of the paths of our dummy consistent attribution path method (DCAPM) and IG with a straight line path and a zero baseline $\mathbf{x}'_{\text{zero}}$.** The target class is the 'French Horn'. The first row represents the trajectories of function output values. The second row shows the path features sampled from four states ($t$) (indicated by the black points in the logit graphs). The third row represents the infinitesimal contributions to be aggregated of the corresponding states, computed by multiplying the difference in path features between one step and the partial derivatives. Our method clearly localizes target-class-relevant features by the path function. It nullifies class-irrelevant features (dummies) in the attribution results until reaching out the predefined baseline $\mathbf{x}'$ or termination conditions. In contrast, IG represents noisy attributions. We provide the detailed optimization progress results in Appendix C.8.

values to dummy features (Figures 1 and 13 in Appendix C.10). This violates the Dummy Consistency axiom by distorting the attributions of the true signal features.

To resolve these axiomatic violations, our method constructs a valid attribution path through a two-stage process. First, we use a gradient-based optimization to find a path that separates dummy and signal features. Second, we introduce a novel mechanism to nullify spurious feature interactions along this path, ensuring a consistent and fair attribution allocation.

### 3.1 PROCESS 1: PATH OPTIMIZATION FOR DUMMY FEATURE SELECTION

We reformulate the task of identifying dummy features as a search for an optimal attribution path. To make this search tractable, we optimize a low-resolution mask $\mathbf{m}^{(t)} \in [0, 1]^{h \times w}$, where $t$ denotes the optimization step. This mask is then upsampled to the full input resolution $M^{(t)} \in [0, 1]^{H \times W}$ ($h \ll H, w \ll W$) via a differentiable bilinear interpolation function, $u(\cdot)$, such that $M^{(t)} = u(\mathbf{m}^{(t)})$. This formulation allows the mask to take on continuous values while significantly reducing the number of trainable parameters. The optimization of $\mathbf{m}$ is made possible by backpropagating through $u(\cdot)$, which involves its Jacobian matrix $\mathcal{W} = \frac{\partial u(\mathbf{m}^{(t)})}{\partial \mathbf{m}^{(t)}}$. The mask is initialized with values of 1 to indicate dummy features and optimized toward 0 in non-dummy regions. To selectively preserve only the dummy features, our optimization objective is defined by the following loss function $\mathcal{L}$:

$$\mathcal{L}(\mathbf{x}, \mathbf{x}', \mathbf{m}^{(t)}) = f(\mathbf{x} \odot u(\mathbf{m}^{(t)}) + \mathbf{x}' \odot (1 - u(\mathbf{m}^{(t)}))) + \lambda |1 - u(\mathbf{m}^{(t)})|_1, \tag{3}$$

where $\odot$ denotes the Hadamard product, and $\lambda$ is a scaling factor for the L1 regularization term to control the rate of feature removal and to encourage sparsity in the resulting mask. The first term generates a path that fades out the relevant feature first. Based on the optimization problem $\min_{\mathbf{m}^{(t)}} \mathcal{L}(\mathbf{x}, \mathbf{x}', \mathbf{m}^{(t)})$, the input state at any step $t$ along the path is defined as:

$$\gamma(t) := \mathbf{x} \odot u(\mathbf{m}^{(t)}) + \mathbf{x}' \odot (1 - u(\mathbf{m}^{(t)})), \quad 0 \le t < \hat{t}, \tag{4}$$

and the mask is updated as $\mathbf{m}^{(t+1)} = \mathbf{m}^{(t)} - \eta \cdot \frac{\partial \mathcal{L}(\mathbf{x}, \mathbf{x}', \mathbf{m}^{(t)})}{\partial \mathbf{m}^{(t)}}$, with learning rate $\eta$. Unlike prior methods that rely on a predefined reference point, our optimization process generates a dynamic path

evolving from the input $\mathbf{x}$ to a per-sample baseline $\mathbf{x}_{\text{dum}}$, determined algorithmically and composed of features identified as dummies at termination of path.

To ensure the optimization process is principled, we introduce two key constraints. First, to be consistent with the Aumann-Shapley framework, the path is constrained by non-negativity, $f(\gamma(t)) \geq 0$, and $\epsilon$-monotonicity, $f(\gamma(t)) \geq f(\gamma(t+1)) + \epsilon$, terminating if either is violated. Second, to address the shattered gradient phenomenon, we define a feature $i$ as a dummy if its gradient magnitude with respect to the mask, $|\partial \mathcal{L}/\partial \gamma_i(t)|$, falls below a threshold $\tau_g$. In practice, the method is robust to $\tau_g$ variations in $[0.001, 0.2]$, as shown in the ablation study (Appendix E.2).

### 3.2 PROCESS 2: NULLIFYING DUMMY INTERACTIONS FOR DUMMY CONSISTENCY

A primary contribution of our work is a mechanism to enforce the Dummy Consistency axiom during the path optimization process. This axiom posits that irrelevant features at one step should not spuriously acquire attribution due to changes in other features.

To formalize this, we first define a relevant feature at any step $t-1$ as any feature whose gradient magnitude exceeds a threshold $\tau_g$. The set of all such features is denoted by $S^{(t-1)}$:

$$S^{(t-1)} := \left\{ i \in \{1, \cdots, h \times w\} \mid \left| \frac{\partial \mathcal{L}(\mathbf{x}, \mathbf{x}', \mathbf{m}^{(t-1)})}{\partial \mathbf{m}_i^{(t-1)}} \right| > \tau_g \right\}. \tag{5}$$

The core problem arises from spurious interactions between signal and dummy features. An update to the signal features $i \in S^{(t-1)}$ along the path from $\gamma(t-1)$ to $\gamma(t)$ can induce a non-zero gradient on a feature $j \notin S^{(t-1)}$ that was previously a dummy, which should be zero. We can quantify this spurious interaction as the finite difference of the partial derivatives of the dummy feature $j$ across the optimization step:

$$\mathcal{I}_{i,j}(f, \gamma, t) = \frac{\partial \mathcal{L}(\mathbf{x}, \mathbf{x}', \mathbf{m}^{(t)})}{\partial \mathbf{m}_j^{(t-1)}} - \frac{\partial \mathcal{L}(\mathbf{x}, \mathbf{x}', \mathbf{m}^{(t-1)})}{\partial \mathbf{m}_j^{(t)}}, \quad \forall i \in S^{(t-1)}, \forall j \notin S^{(t-1)}. \tag{6}$$

To satisfy the Dummy Consistency axiom, this interaction term $\mathcal{I}_{i,j}$ must be zero. We enforce this by constraining the optimization to a set $C^{(t)}$ that contains only features which are not dummy in one step interval, which is defined as:

$$C^{(t)} = \left\{ i \in \mathscr{M} \mid i \in S^{(t-1)} \text{ and } i \in S^{(t)} \right\}, \quad \mathscr{M} = \{1, ..., h \times w\}. \tag{7}$$

By restricting the mask update to this set, we explicitly prevent the optimization from inducing gradients on dummy features. The corresponding update rule for the mask is thus given by[2]:

$$\mathbf{m}_i^{(t)} = \mathbf{m}_i^{(t-1)} - \eta \cdot \text{sign}\left( \frac{\partial \mathcal{L}(\mathbf{x}, \mathbf{x}', \mathbf{m}^{(t-1)})}{\partial \mathbf{m}_i^{(t-1)}} \right), \quad \forall i \in C^{(t)}. \tag{8}$$

### 3.3 DUMMY CONSISTENT ATTRIBUTION PATH METHOD (DCAPM)

Building on the path function (Eq. 4) and mask update rule (Eq. 8), we derive the derivative of $\gamma(t)$ with respect to the iteration step $t$ using the chain rule:

$$\begin{aligned}
\frac{\partial \gamma(t)}{\partial t} &= \mathbf{x} \odot \frac{\partial u(\mathbf{m}^{(t)})}{\partial t} + \mathbf{x}' \odot \left( -\frac{\partial u(\mathbf{m}^{(t)})}{\partial t} \right) = (\mathbf{x} - \mathbf{x}') \odot \frac{\partial u(\mathbf{m}^{(t)})}{\partial t} \\
&= (\mathbf{x} - \mathbf{x}') \odot \frac{\partial u(\mathbf{m}^{(t)})}{\partial \mathbf{m}^{(t)}} \cdot \frac{\partial \mathbf{m}^{(t)}}{\partial t} = (\mathbf{x} - \mathbf{x}') \odot \mathcal{W} \cdot \left[ \eta \cdot \text{sign}\left( \frac{\partial \mathcal{L}(\mathbf{x}, \mathbf{x}', \mathbf{m}^{(t)})}{\partial \mathbf{m}^{(t)}} \right) \right].
\end{aligned} \tag{9}$$

---

[2]Note that in practice we add a $\text{sign}(\cdot)$ for stable gradient descent updates of the mask, inspired by Projected Gradient Descent (PGD) (Madry et al., 2018). We additionally employ a cyclical learning rate schedule (Smith, 2017), to mitigate local minima issues (see Appendix B for full algorithmic details).

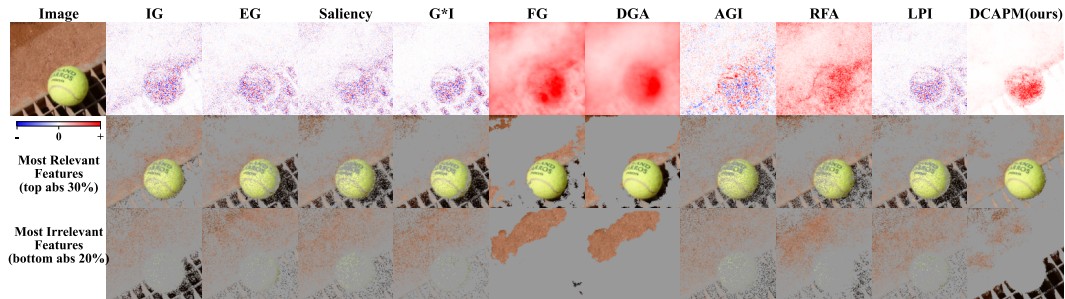

Figure 3: **Qualitative comparison of benchmark attribution methods and our proposed approach for the class 'Tennis ball' from the ImageNet validation set.** (Top Row) Full attribution maps. (Middle Row) The top 30% of the absolute attributions, highlighting the most relevant features. (Bottom Row) The bottom 20% of attributions, representing the least relevant features.

By incorporating the derived derivative from Eq. 9 and the constructed path function from Eq. 4 into the original path-based attribution framework (Eq. 1), we derive the proposed Dummy Consistent Attribution Path Method (DCAPM) as follows:

$$\mathcal{A}_i^\gamma(\mathbf{x}, \mathbf{x}'; f) = \eta \sum_{t=0}^{\hat{t}-1} \big(\gamma_i(t) - \gamma_i(t+1)\big) \cdot \frac{\partial f(\gamma(t))}{\partial \gamma_i(t)} \cdot \mathcal{W}_i \cdot \text{sign}\left(\frac{\partial \mathcal{L}(\mathbf{x}, \mathbf{x}', \mathbf{m}^{(t)})}{\partial \mathbf{m}^{(t)}}\right), \quad (10)$$

where the term $\mathcal{W}_i$ denotes a scalar weight associated with the $i$-th feature. During optimization, we aggregate the partial derivatives of valid path features, weighted by $\mathcal{W} \cdot \partial \mathcal{L}/\partial \mathbf{m}^{(t)}$ along the path until the termination conditions. Note that the per-step path displacement term $\gamma_i(t) - \gamma_i(t+1)$ replaces the total difference term $\mathbf{x} - \mathbf{x}'$ to allow the discrete implementation from the continuous integral of the path method in Eq 1. By constructing a path that explicitly nullifies dummy interactions, DCAPM provides attributions that satisfy the Dummy Consistency axiom. The detailed algorithm, proof of axiom satisfaction, and complete derivation of DCAPM are provided in Appendices B, I, and J.

## 4 DUMMY CONSISTENCY EVALUATION METRIC

Grounded in Definition 4 of the Dummy Consistency axiom, we propose a novel metric to quantitatively assess the adherence of an attribution method to this axiom. Since dummy partial derivatives are rarely exactly zero in deep neural networks (DNNs), our metric measures the practical deviation by computing the Mean Squared Error (MSE) between attributions of signal features before and after removing estimated dummy features:

$$\text{Dummy-Consistency} = \frac{1}{|N \setminus \widetilde{M}_{\text{dum}}|} \sum_{i \in N \setminus \widetilde{M}_{\text{dum}}} \|\widetilde{\mathcal{A}}_i(\mathbf{x}, \mathbf{x}', f) - \widetilde{\mathcal{A}}_i(\mathbf{x} \odot (1 - \widetilde{M}_{\text{dum}}), \mathbf{x}', f)\|^2, \quad (11)$$

where $\widetilde{M}_{\text{dum}} = \{i \in N \mid \widetilde{\mathcal{A}}_i < \tau_{dum}\}$ is the estimated dummy feature set with threshold $\tau_{dum}$ = 0.1, as applied in our experiments, and $\widetilde{\mathcal{A}}$ represents min-max normalized absolute attribution values to mitigate scale variance across methods and samples. The element-wise product $\odot$ masks dummy features, and a lower score indicates higher dummy consistency, reflecting stable signal feature attributions unaffected by irrelevant dummies.

## 5 EXPERIMENTAL RESULTS

We empirically demonstrate the identified phenomenon in a deep learning classification task and assess the effectiveness of our proposed method. We utilize pre-trained image classifiers, VGG16 (Simonyan & Zisserman, 2015), InceptionV3 (Szegedy et al., 2016), and ResNet18 (He et al., 2016), all trained on the ImageNet dataset (Deng et al., 2009), with an $8 \times 8$ spatial mask resolution employed for optimization. Detailed descriptions of the benchmark methods, experimental configurations, and implementation details are provided in Appendices D and F.

Table 1: **Quantitative results of sensitivity and consistency evaluation scores across different attribution methods on the three models: VGG16, InceptionV3 (IncV3), and ResNet18 (RN18).** Higher score is better for DiffID (↑) and lower is better for Non-Sensitivity (↓) and Dummy Consistency (↓). The **red** and **blue** denote the best and second-best scores in each column, respectively.

| | DiffID (↑) | | | Non-Sensitivity (↓) | | | Dum-Consistency (↓) | | |
|---|---|---|---|---|---|---|---|---|---|
| | VGG16 | IncV3 | RN18 | VGG16 | IncV3 | RN18 | VGG16 | IncV3 | RN18 |
| IG | .012 ±.01 | .061 ±.21 | .025 ±.08 | .0012 ±.00 | .0018 ±.00 | .0018 ±.00 | **.0010 ±.00** | .201 ±.06 | .203 ±.05 |
| EG | .014 ±.02 | .066 ±.38 | .025 ±.10 | .0012 ±.00 | .0018 ±.00 | .0018 ±.00 | .0012 ±.00 | .195 ±.04 | .190 ±.04 |
| Saliency | .010 ±.01 | .065 ±.36 | .023 ±.08 | .0012 ±.00 | .0018 ±.00 | .0018 ±.00 | .0013 ±.00 | .183 ±.04 | .175 ±.04 |
| I×G | .011 ±.01 | .068 ±.35 | .026 ±.10 | .0012 ±.00 | .0018 ±.00 | .0018 ±.00 | .0008 ±.00 | .185 ±.05 | .168 ±.04 |
| FG | .041 ±.04 | .083 ±.31 | .059 ±.24 | .0011 ±.00 | **.0009 ±.00** | **.0009 ±.00** | .0114 ±.01 | .265 ±.06 | .299 ±.07 |
| DGA | **.046 ±.05** | .090 ±.38 | **.061 ±.20** | .0011 ±.00 | .0013 ±.00 | .0014 ±.00 | .0703 ±.04 | .225 ±.05 | .256 ±.06 |
| AGI | .019 ±.02 | .059 ±.19 | .028 ±.09 | .0012 ±.00 | .0018 ±.00 | .0018 ±.00 | .0024 ±.00 | .200 ±.05 | .198 ±.05 |
| RFA | **.038 ±.05** | **.092 ±.39** | .049 ±.16 | **.0010 ±.00** | .0012 ±.00 | .0012 ±.00 | .0037 ±.00 | **.011 ±.01** | **.005 ±.00** |
| LPI | .013 ±.01 | .058 ±.25 | .025 ±.09 | .0012 ±.00 | .0018 ±.00 | .0018 ±.00 | .0012 ±.00 | .196 ±.04 | .189 ±.04 |
| **DCAPM** | .036 ±.06 | **.098 ±.64** | **.059 ±.25** | **.0002 ±.00** | **.0003 ±.00** | **.0003 ±0.00** | **.0004 ±.00** | **.0001 ±.00** | **.00002 ±.00** |

## 5.1 QUALITATIVE COMPARISON

As illustrated in Figure 3, our proposed method effectively nullifies interactions involving dummy features, thereby producing less noisy attributions in irrelevant regions. This improvement is particularly evident when visualizing the top 30% of features with the highest absolute attribution values; our method distinctly highlights the target class, Tennis ball, with minimal background noise. Furthermore, when visualizing the bottom 20% of features, our method achieves a clearer distinction by reducing unstable gradient signals in background regions that arise from shattered gradients. Additional qualitative examples are provided in Appendix C.1.

## 5.2 ENFORCING DUMMY CONSISTENCY IMPROVES STANDARD SENSITIVITY METRICS

We evaluate our method using two sensitivity metrics: Non-Sensitivity (Nguyen & Martínez, 2020) and DiffID (Yang et al., 2023), which measures the difference between Insertion and Deletion game scores (Petsiuk et al., 2018). As shown in Table 1, DCAPM not only achieves the best performance in Non-Sensitivity but also performs competitively with the best refinement methods, such as DGA, on DiffID. These results indicate that enforcing dummy consistency enhances attribution robustness without compromising sensitivity, as reflected in improved performance on these orthogonal metrics. Additional quantitative results on the Flowers and Oxford-IIIT Pet datasets are provided in Appendix C.7. The additional impact of nullifying dummy interactions is verified in Appendix E.1.

## 5.3 SCALABILITY AND APPLICATION TO TEXT DOMAIN

To verify the robustness of DCAPM across different modalities and architectures, we analyze the entropy of attribution distributions across 500 test samples using a BERT-Large model fine-tuned on the IMDb dataset. As shown in Figure 4, DCAPM achieves an average entropy of 4.13, significantly lower than the entropy of IG at 5.70. The results indicate that DCAPM produces more sparse attributions, concentrating importance on key semantic tokens rather than diffusing it across irrelevant ones. This demonstrates that the effectiveness of DCAPM extends beyond vision tasks to different modalities and complex architectures. Further details on experimental setup and qualitative examples are provided in Appendix C.3.

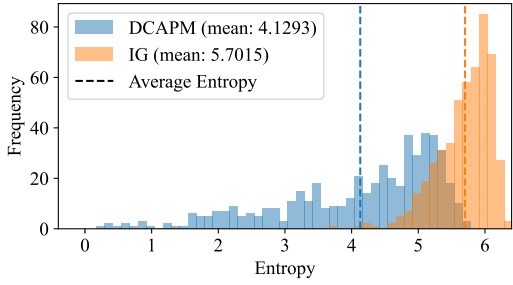

Figure 4: **Entropy analysis to compare IG and our proposed method, DCAPM,** using the **BERT-Large** sentiment classifier model on randomly sampled 500 instances from the IMDb test dataset.

### 5.4 VALIDATION OF THE PROPOSED DUMMY CONSISTENCY METRIC

As shown in Table 1, our method achieves the best score on the Dummy Consistency metric, which measures attribution stability when dummy features are introduced. The relatively poor performance of methods like FG and DGA can be attributed to inherent biases that assign non-zero attribution to irrelevant features. Notably, masking mechanism of DGA improves its score over its base method, FG, supporting the validity of our proposed metrics. Results from the correlation analysis, detailed in Appendix C.11, indicate that the metric is statistically distinct from existing sensitivity measures (Insertion/Deletion, DiffID, Non-Sensitivity), highlighting a novel aspect of attribution quality.

## 6 RELATED WORK

**Enhancing Consistency in Path-Based Attribution Methods.** Several studies have addressed the counterintuitive and inconsistent attribution results of path-based methods like Integrated Gradients (IG) (Sundararajan et al., 2017). FullGrad (FG) (Srinivas & Fleuret, 2019) mitigates issues arising from traversing different linear regions in piecewise-linear networks by aggregating gradients across them. However, as the difference in gradients approximates the Shapley interaction index in DNNs (see Chang et al., 2025, Eq. 10), we interpret these varying gradients as the contributions of feature interactions, which our method aims to control directly. Other works have approached this problem through refinement. Distilled Gradient Aggregation (DGA) (Jeon et al., 2022) is a heuristic-based refinement approach that iteratively removes attribution from irrelevant features. Similarly, Recalibrating Feature Attributions (RFA) (Yang et al., 2022) addresses the cancellation of positive and negative attributions along an integration path. However, such cancellation effects are naturally avoided if the foundational Aumann-Shapley assumptions, such as Monotonicity, are met, which is a key design principle of our proposed method. Alternative strategies have focused on path and baseline selection. LPI (Yang et al., 2023) proposes a clustering method in the activation space of a single layer to select valid baselines. However, its reliance on a single layer does not fully capture the piecewise-linear structure of the model. In contrast, our method leverages model gradients, which are deterministic within each linear region. Adversarial Gradient Integration (AGI) (Pan et al., 2021) searches for paths by optimizing toward adversarial targets. While these studies improve fidelity performance, they approach the issue of consistency from empirical or intuitive perspectives and do not directly address violations of the Dummy Consistency axiom in rectifier networks. In contrast, our work is rigorously grounded in the Dummy Player and Dummy Consistency axioms, which facilitate the systematic design of both our proposed algorithm and its corresponding evaluation metric.

**Evaluating Attributions for Consistency.** In existing literature, the evaluation of *consistency* often pertains to the stability of function outputs (Dasgupta et al., 2022) or the robustness of explanations to input perturbations (Alvarez Melis & Jaakkola, 2018; Agarwal et al., 2022; Montavon et al., 2018). One relevant metric is *Non-Sensitivity* (Nguyen & Martínez, 2020), which assesses the insensitivity of attributions to the introduction of dummy features. This metric directly aligns with the Dummy Player Axiom, which states that the attribution for a feature on which the model is not functionally dependent must be zero. However, to our knowledge, metrics that directly evaluate satisfaction of the Dummy Consistency axiom remain underexplored. This axiom, which dictates that the attributions of signal features should not be altered by the presence of a dummy feature. To address this gap, we introduce both a method designed to satisfy this axiom and a novel metric to explicitly measure it.

## 7 CONCLUSION

We introduced the *Dummy Consistent Attribution Path Method* (DCAPM), an optimization-based path method designed to eliminate the influence of irrelevant features. Through the lens of the Shapley interaction index, DCAPM dynamically constructs an integration path that nullifies spurious attributions from dummy feature interactions, ensuring zero attribution for irrelevant features at every step by enforcing dummy consistency. Extensive experimental comparisons with benchmark methods demonstrate the effectiveness of DCAPM, showing improved performance on sensitivity metrics by enforcing dummy consistency, thus highlighting the robustness and reliability of our proposed approach. We refer readers to Appendix K.1 for a discussion of the limitations and future directions.

**Reproducibility Statement.** For reproducibility, we provide the pseudocode for our proposed method and metric in Appendix B, with the full Python implementation available in Appendix F. Comprehensive experimental details, including hyperparameter settings, benchmark model implementations, and computational resources, are detailed in Appendices D and G.

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
