# Appendix

## Table of Contents

## A   NOTATION

Table 2 summarizes the mathematical notations used throughout this paper, defining key symbols for our attribution framework and optimization-based path method.

Table 2: **Table of notation**.

| Notation | Description |
| --- | --- |
| $N$ | The set of all features |
| $N \setminus \{i\}$ | The set of all features except for feature $i$ |
| $\mathcal{X}(N)$ | The space of all possible inputs with $N$ features |
| $\mathbf{x} \in \mathbb{R}^{|N|}$ | An $|N|$-dimensional input vector |
| $x_i$ | The value of the $i$-th feature of the input vector $\mathbf{x}$ |
| $\mathbf{x}_{-i}$ | The input vector $\mathbf{x}$ with the $i$-th feature excluded |
| $f(\mathbf{x}_{-i})$ | The function value of the remaining features $\mathbf{x}_{-i}$ with respect to the dummy feature $i$ |
| $f^{R_i^{\mathbf{x}}}$ | The reduced function on the given function $f$ with respect to the remaining features $\mathbf{x}_{-i}$ |
| $\mathbf{x}'$ | A reference point (the pseudo end point of a path) |
| $\mathbf{x}_{\text{dum}}$ | An estimated baseline point (the end point of the path) composed of dummy features |
| $\mathcal{A}^{\gamma}(\mathbf{x}, \mathbf{x}'; f)$ | An attribution method of feature $\mathbf{x}$ to the prediction of model $f$ with a path function $\gamma$ |
| $\mathcal{A}_i^{\gamma}(\mathbf{x}, \mathbf{x}'; f)$ | The $i$-th attribution value of feature $\mathbf{x}$ to the prediction of model |
| $\gamma(t)$ | The path function at step $t$ between input $\mathbf{x}$ and reference $\mathbf{x}'$ |
| $t, \hat{t}$ | The discrete time step of the path integration and its final step. |
| $\mathbf{m}^{(t)}$ | The mask parameter at optimization step $t$, typically with smaller dimensions than the input |
| $u(\cdot)$ | The upsampling function that maps the low-resolution mask $\mathbf{m}$ to the input resolution |
| $M^{(t)}$ | The upsampled mask $u(\mathbf{m}^{(t)})$ |
| $\widetilde{M}_{\text{dum}}$ | The estimated set of dummy features |
| $\mathcal{L}$ | The objective function for dummy feature selection |
| $\mathcal{W}$ | The Jacobian matrix of the upsampling function $u(\cdot)$ |
| $\lambda$ | The scaling factor for the L1 regularization term |
| $\tau_g$ | The threshold for identifying dummy features and dummy interactions |
| $\tau_{dum}$ | The threshold for identifying dummy feature set |
| $\eta$ | The learning rate (w/o cyclical learning rate) |
| $\eta^{(t)}$ | The cyclical learning rate at iteration $t$ |
| $C^{(t)}$ | The active set of non-dummy features at iteration $t$ |
| $\mathcal{I}_{i,j}$ | The Shapley interaction value between features $i$ and $j$ |

## B   ALGORITHM DETAILS

The proposed method is detailed in Algorithm 1. For notational simplicity, we use $\nabla_{\mathbf{m}^{(t)}} \mathcal{L}$ to denote the gradient $d\mathcal{L}(\mathbf{x}, \mathbf{x}', \mathbf{m}^{(t)})/d\mathbf{m}^{(t)}$. A clipping operation is applied after each update to constrain the mask values within the valid interval $[0, 1]$. The optimization process halts when any of the following

three termination criteria are met: **(1) Non-Negativity Condition:** The condition terminates the process if the function output $f(\mathbf{x}^{(t)})$ is no longer positive (Line 6), ensuring the path has reached a region of non-positive contribution. **(2) Mask Convergence (Saturation):** The condition checks for the convergence of the mask itself. It halts the optimization if the change between consecutive mask updates falls below a predefined threshold, using a standard check that accounts for both absolute and relative tolerance (Line 19). **(3) $\epsilon$-Monotonicity Safeguard:** The third criterion is a safeguard that enforces $\epsilon$-monotonicity along the path. It terminates the process if a step unexpectedly increases the function value by more than $\epsilon$ (Line 20), preventing the accumulation of attribution from non-monotonic segments.

---

**Algorithm 1 Dummy Consistent Attribution Path Method**

---

1: **Input:** Input $\mathbf{x}$
2: **Output:** Attribution $\mathcal{A}(\mathbf{x})$
3: **Parameter:** Mask $\mathbf{m}$
4: **Definitions:** Global reference point $\mathbf{x}'$; dummy feature threshold $\tau_g$; small value $\epsilon$; the base learning rate $\eta_{\min}$; the maximum learning rate $\eta_{\max}$; the number of step size $T_{\text{cycle}}$, and the initial learning rate $\eta^0$; path state and iteration step $t$; maximum number of iterations $T_{fin}$; upsampling function $u : \mathbf{m} \in [0,1]^{h \times w} \mapsto \mathcal{W}\mathbf{m} \in [0,1]^{H \times W}$; bilinear interpolation weight matrix $\mathcal{W}$.
5: **Initialize:** $\mathbf{m}^{(0)} \leftarrow \mathbf{1}^{h \times w}$; $x_i' \leftarrow \min_{\mathbf{x} \in D} x_i$; $\mathcal{A} \leftarrow \mathbf{0}^{H \times W}$; $C \leftarrow \emptyset$; $\tau_g \leftarrow 0.1$, $\epsilon \leftarrow 10^{-4}$; $\gamma(0) \leftarrow \mathbf{x} \odot u(\mathbf{m}^{(0)}) + \mathbf{x}' \odot (1 - u(\mathbf{m}^{(0)}))$; $\nabla_{\mathbf{m}^{(0)}} \mathcal{L} \leftarrow \frac{d\mathcal{L}(\mathbf{x}, \mathbf{x}', \mathbf{m}^{(0)})}{d\mathbf{m}^{(0)}}$; $t \leftarrow 0$; $T_{fin} \leftarrow 600$; $\eta_{\min} \leftarrow 10^{-3}$; $\eta_{\max} \leftarrow 0.01$; $T_{\text{cycle}} \leftarrow 10$; $\eta^{(0)} \leftarrow 10^{-3}$;.
6: **while** $f(\gamma(t)) > 0$ **and** $t < T_{fin}$ **do**                    ▷ Non-negativity condition
7:     $\nabla_{\mathbf{m}^{(t)}} \mathcal{L} \leftarrow \dfrac{d\mathcal{L}(\mathbf{x}, \mathbf{x}', \mathbf{m}^{(t)})}{d\mathbf{m}^{(t)}}$
8:     $q_\tau = \text{sort}(|\nabla_{\mathbf{m}^{(t)}} \mathcal{L}|)[\lfloor \tau_g \cdot (h \times w - 1) \rfloor]$          ▷ $\tau_g$-quantile value (ascending order)
9:     **if** $t > 1$ **then**
10:         $C = \{ i \mid |\nabla_{m_i^{(t-1)}} \mathcal{L}| > q_\tau \text{ and } \nabla_{m_i^{(t)}} \mathcal{L} > \tau_g \}$          ▷ Masking dummy interactions
11:     **else**
12:         $C = \{ i \mid \nabla_{m_i^{(t)}} \mathcal{L} > \tau_g \}$
13:     **end if**
14:     **for** $i$ **in** $C$ **do**
15:         $m_i^{(t+1)} = m_i^{(t)} - \eta^{(t)} \cdot \text{sign}\left(\nabla_{m_i^{(t)}} \mathcal{L}\right)$          ▷ Dummy mask optimization
16:         $m_i^{(t+1)} = \text{Clip}(m_i^{(t+1)}, 0, 1)$
17:     **end for**
18:     $\gamma(t+1) \leftarrow \mathbf{x} \odot u(\mathbf{m}^{(t+1)}) + \mathbf{x}' \odot (1 - u(\mathbf{m}^{(t+1)}))$
19:     **if** $|m_i^{(t)} - m_i^{(t+1)}| \leq \epsilon \cdot (1 + |m_i^{(t+1)}|) \quad \forall i$          ▷ Trivial difference condition
20:         **or** $f(\gamma(t)) - f(\gamma(t+1)) < \epsilon$ **then**          ▷ $\epsilon$-Monotonicity condition
21:         **Terminate**
22:     **end if**
23:     $\mathcal{A}_i += (\gamma_i(t) - \gamma_i(t+1)) \cdot \dfrac{\partial f(\gamma(t+1))}{\partial \gamma_i(t+1)} \cdot \mathcal{W}_i \cdot \eta^{(t)} \cdot \nabla_{\mathbf{m}^{(t+1)}} \mathcal{L}$     ▷ Aggregate attributions
24:     $\eta^{(t+1)} = \eta_{\min} + (\eta_{\max} - \eta_{\min}) \cdot \max(0, 1 - |\frac{t - T_{\text{cycle}}}{T_{\text{cycle}}}|)$          ▷ Cyclical learning rate
25:     $t += 1$
26: **end while**
27: **return** $\mathcal{A} \leftarrow \mathcal{A}/t$

---

# C ADDITIONAL EXPERIMENTS

## C.1 QUALITATIVE EVALUATION ON THE VGG16 WITH IMAGENET

A qualitative evaluation on the ImageNet validation set visually demonstrates our method's superior ability to distinguish between signal and noise. As shown in Figures 5 and 6, our approach produces sparse attribution maps. This stands in contrast to existing axiomatic methods, which yield diffuse and noisy results, and provides a cleaner separation of relevant and irrelevant features.

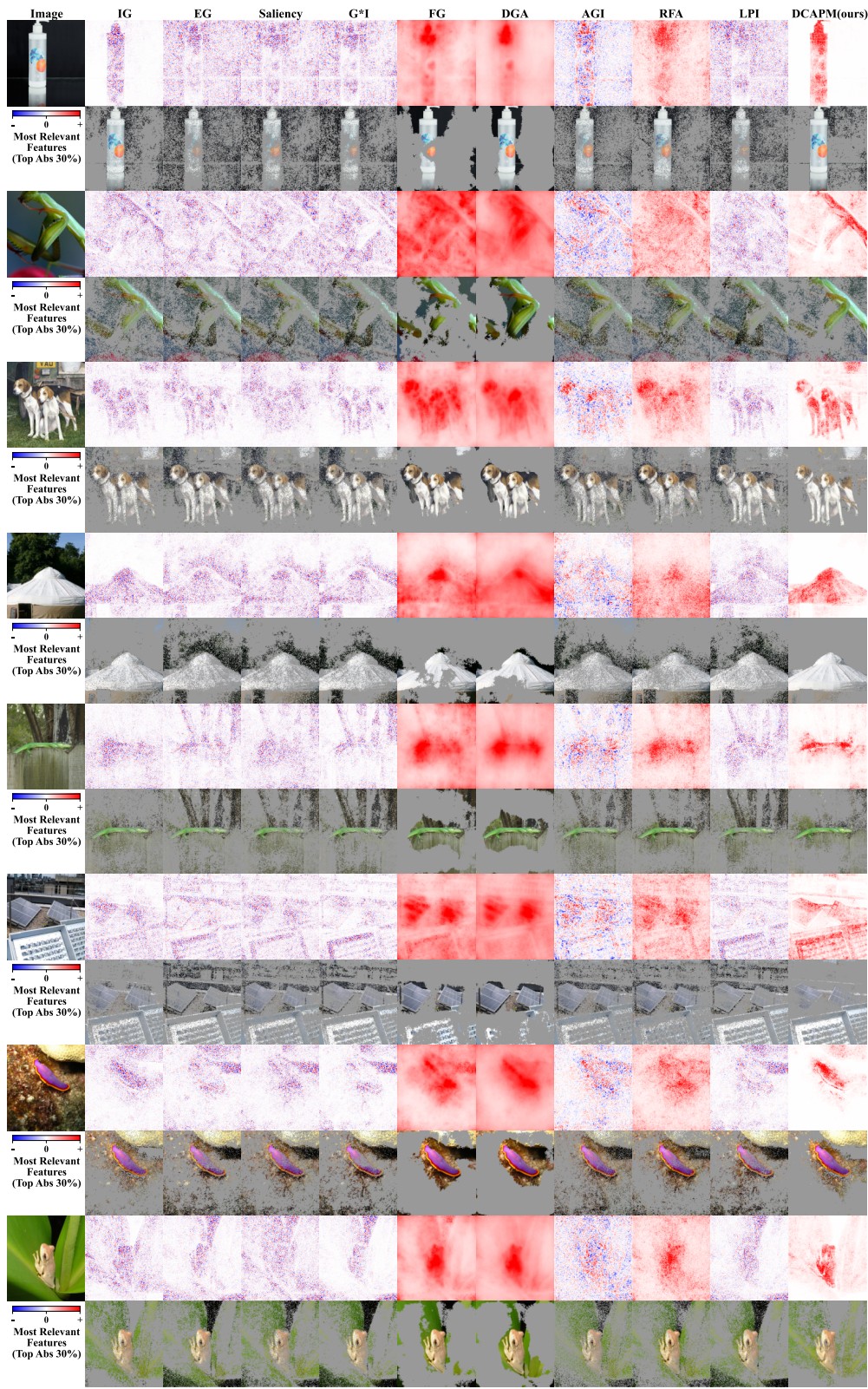

Figure 5: **Qualitative comparison of the most salient features (top 30%).** The top row for each method shows the full attribution map, while the bottom row highlights only the top 30% of absolute attribution values. Our method's top attributions are sharply focused on the foreground object, whereas competing methods include significant background noise.

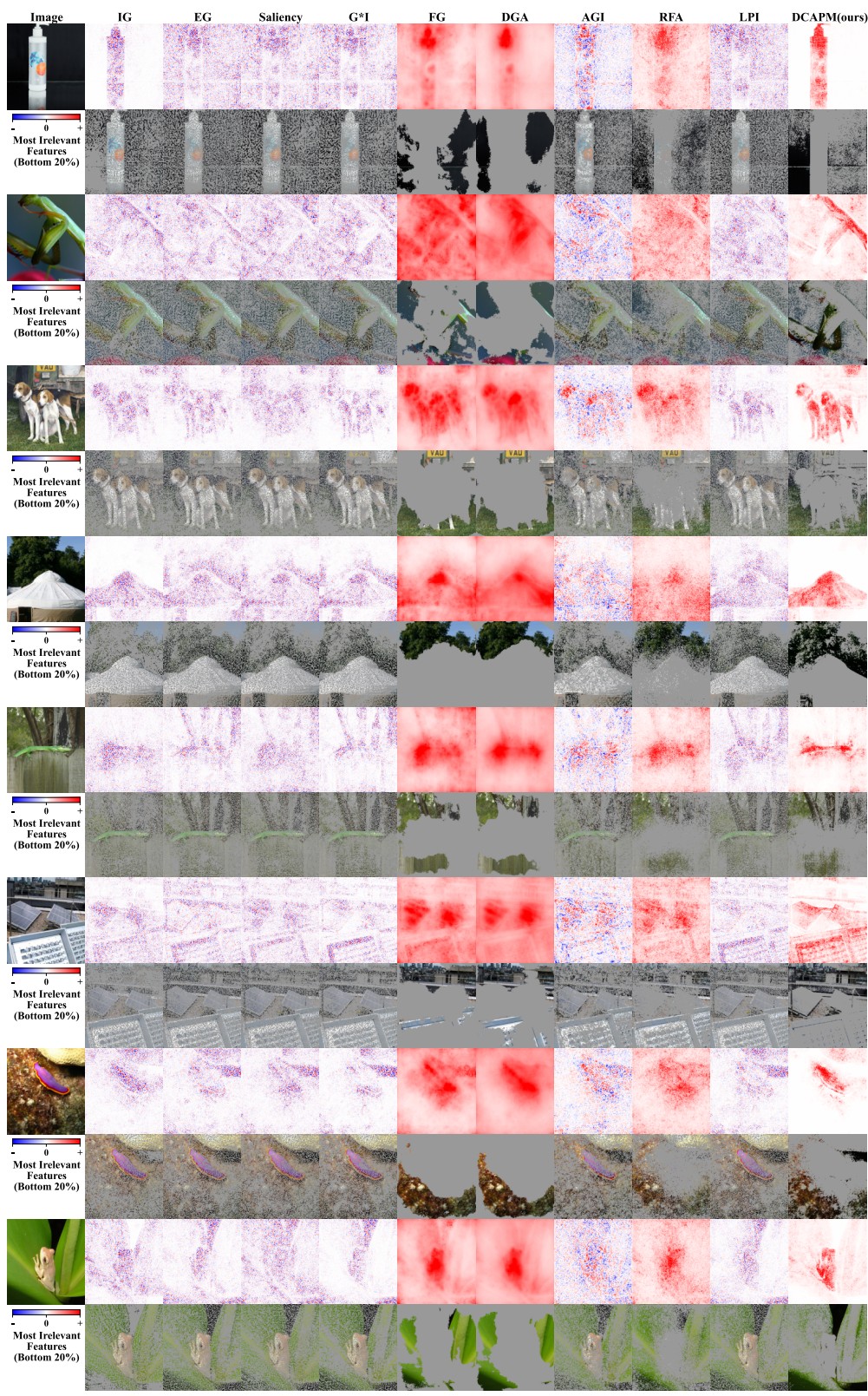

Figure 6: **Qualitative comparison of the least salient features (bottom 20%).** This visualization highlights the features identified as most irrelevant (i.e., potential dummy features). Our method correctly identifies only background regions as irrelevant, while competing methods erroneously include noise or even parts of the foreground object in their bottom attributions.

## C.2 ADDITIONAL EXPERIMENT OF MOTIVATION FIGURE 1 ON TWOMOONS DATASET

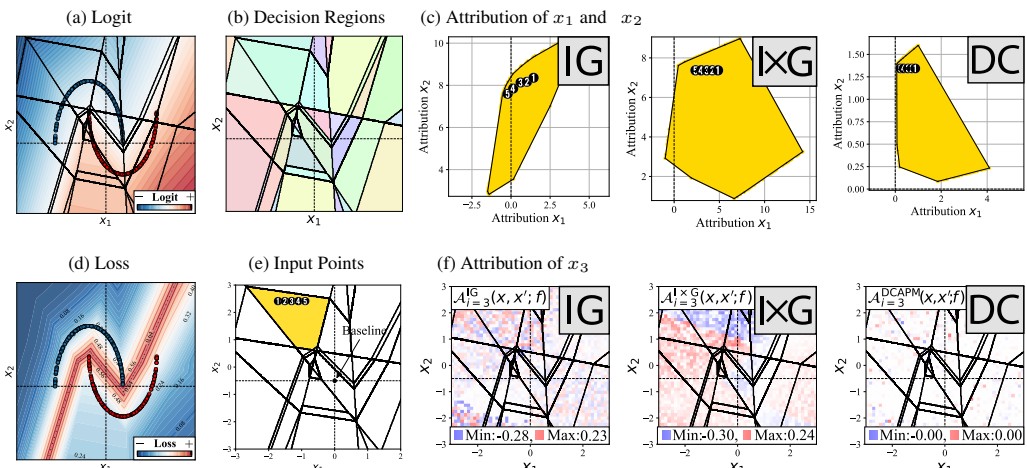

Figure 7: **Additional Experiment of Figure 1 on TwoMoons.** A noise feature $x_3$ is added to the 2D binary classification task. **(a, b, d, e) Model Behavior:** Logit/loss outputs and decision boundaries for signal features $x_1$ and $x_2$. Points ❶–❺ (yellow region) correspond to a single-feature-axis-shift scenario. **(c) Dummy Consistency Violation:** IG shows the attributions for $x_1$ and $x_2$ are distorted by $x_3$. However, DCAPM (Ours) and non-path-based I×G remain undistorted. **(f) Dummy Player Violation:** IG and I×G assign non-zero attribution to the irrelevant $x_3$, whereas DCAPM correctly assigns it almost zero attribution.

## C.3 QUALITATIVE AND QUANTITATIVE EVALUATION ON THE IMDB DATASET

To assess the generalizability of our method beyond CNNs, we conducted a comprehensive evaluation on Transformer-based architectures. We compare our method, DCAPM, against Integrated Gradients (IG) on BERT models fine-tuned for sentiment classification on the IMDb movie review dataset.

**Experimental Setup** For qualitative analysis, we visualize the L2 norm of attribution values, where greener shades indicate higher input importance. To ensure a fair comparison, the visualized samples (#8 and #18) were specifically chosen from the overlapping region of the two methods' entropy distributions (Figure 4), where they exhibit nearly identical entropy values. For quantitative analysis, we measured the average entropy of attributions over the first 500 samples of the IMDb test set. Detailed model and hyperparameter configurations are provided in Appendix D.

**Results and Analysis** As shown in Tables 3 and 4, the qualitative results reveal a clear distinction between the two methods. Our method (DCAPM) produces more focused attributions, concentrating on semantically relevant tokens for a 'Negative' prediction such as "pains", "horrible", and "blame," while effectively suppressing attribution on irrelevant tokens. In contrast, IG yields noisier attributions, often assigning high importance to functionally meaningless tokens like "[CLS]", "[SEP]", and commas. This qualitative finding is supported by our quantitative analysis. As shown in Figure 4, DCAPM achieves a significantly lower average entropy (4.1293) compared to IG (5.7015). This confirms that our method produces sparser and more concentrated attributions, while IG tends to distribute attribution more diffusely across a wider range of tokens, evevn if the BERT architectures consist of GELU activation function. Collectively, these results confirm the robustness and scalability of our method across three key dimensions:

**i) Robustness to Model Depth** Our method's effectiveness is not constrained by model depth. The qualitative results demonstrate that DCAPM provides more semantically meaningful attributions than IG on both the 12-layer BERT-Base and the deeper 24-layer BERT-Large models. This is consistent with our findings on CNNs of varying depths (e.g., VGG16, ResNet18, and InceptionV3), confirming the stability of our approach.

**ii) Generalization to Transformer Architectures**   The experiments on BERT-Base and BERT-Large confirm that our method, initially evaluated on CNNs, generalizes effectively to Transformer-based architectures. In both cases, DCAPM was qualitatively superior to the IG baseline in identifying the input features most relevant to the model's sentiment classification.

**iii) Scalability with Model Size**   DCAPM scales effectively to models of substantially different sizes. The quality of our results on the 334M-parameter BERT-Large model is comparable to that on the 110M-parameter BERT-Base model. This indicates that our method's performance advantage is maintained even as the model size and parameter count increase threefold.

Table 3: **Qualitative comparison of IG and our method (DCAPM) on a sample (#8) from the IMDb test set, using the BERT-Base sentiment classifier.** Attribution maps visualize the L2 norm of attribution values, where greener shades indicate higher input importance.

| Method | Prediction | Entropy | Word Importance |
|--------|-----------|---------|-----------------|
| **IG** | Negative | 5.9269 | #CLS it actually pains me to say it , but this movie was horrible on every level . the blame does not lie entirely with van dam ##me as you can see he tried his best , but let ' s face it , he ' s almost fifty , how much more can you ask of him ? i find it so hard to believe that the same people who put together und ##is ##puted 2 ; arguably the best ( western ) martial arts movie in years , created this . everything from the plot , to the dial ##og , to the editing , to the overall acting was just horribly put together and in ma #SEP |
| **DCAPM** | Negative | 5.3265 | #CLS it actually pains me to say it , but this movie was horrible on every level . the blame does not lie entirely with van dam ##me as you can see he tried his best , but let ' s face it , he ' s almost fifty , how much more can you ask of him ? i find it so hard to believe that the same people who put together und ##is ##puted 2 ; arguably the best ( western ) martial arts movie in years , created this . everything from the plot , to the dial ##og , to the editing , to the overall acting was just horribly put together and in ma #SEP |

Legend:   ■ Zero Importance ■ High Importance

Table 4: **Qualitative comparison of IG and our method (DCAPM) on a sample** (#18) **from the IMDb test set, using the BERT-Large sentiment classifier.** Attribution maps visualize the L2 norm of attribution values, where greener shades indicate higher input importance.

| Method | Prediction | Entropy | Word Importance |
|--------|-----------|---------|-----------------|
| **IG** | Negative | 5.5144 | #CLS i just finished watching this movie and am disappointed to say that i didn ' t enjoy it a bit . it is so slow slow and un ##int ##eres ##ting . this kid from harry potter plays a shy teenager with an rude mother , and then one day the rude mother tells the kid to find a job so that they could accommodate an old guy apparently having no place to live has started to live with his family and therefore the kid goes to work for a old lady . and this old lady who is living all alone teaches him about girls , driving #SEP |
| **DCAPM** | Negative | 5.1440 | #CLS i just finished watching this movie and am disappointed to say that i didn ' t enjoy it a bit . it is so slow slow and un ##int ##eres ##ting . this kid from harry potter plays a shy teenager with an rude mother , and then one day the rude mother tells the kid to find a job so that they could accommodate an old guy apparently having no place to live has started to live with his family and therefore the kid goes to work for a old lady . and this old lady who is living all alone teaches him about girls , driving #SEP |

Legend: ■ Zero Importance ■ High Importance

## C.4 ADDITIONAL QUANTITATIVE EVALUATION ON CONVNEXT

In this section, we quantitatively evaluate our proposed method, DCAPM, against baseline approaches—DGA and Integrated Gradients (IG)—focusing on attribution quality (DiffID) and computational cost. Experiments were conducted using the ConvNeXt-T model (Liu et al., 2022), a modern architecture pretrained on ImageNet-1K with 28M parameters. We test on 500 randomly selected samples from ImageNet with seed 42. DCAPM achieved the best attribution fidelity with a mean DiffID score of $0.4801$, surpassing both DGA ($0.4737$) and IG ($0.2306$). DCAPM offered a superior balance of quality and efficiency, maintaining a reasonable runtime.

Table 5: **Quantitative comparison with baseline methods.** The mean and standard deviation for DiffID scores (higher is better) and runtime in seconds are reported. DCAPM achieves the best DiffID performance with reasonable computational cost.

| Method | DiffID ($\uparrow$) | Runtime (s) ($\downarrow$) |
|---|---|---|
| IG | $0.2306 \pm 0.1245$ | $\mathbf{0.97 \pm 0.41}$ |
| DGA | $0.4737 \pm 0.1989$ | $253.33 \pm 404.29$ |
| **DCAPM (Ours)** | $\mathbf{0.4801 \pm 0.2133}$ | $134.27 \pm 165.51$ |

## C.5 SPURIOUS CORRELATION ANALYSIS VIA DCAPM ATTRIBUTION

In this section, we demonstrate the utility of DCAPM for model debugging. An effective attribution method should faithfully reveal spurious correlations learned by the model (Adebayo et al., 2020; Schramowski et al., 2020).

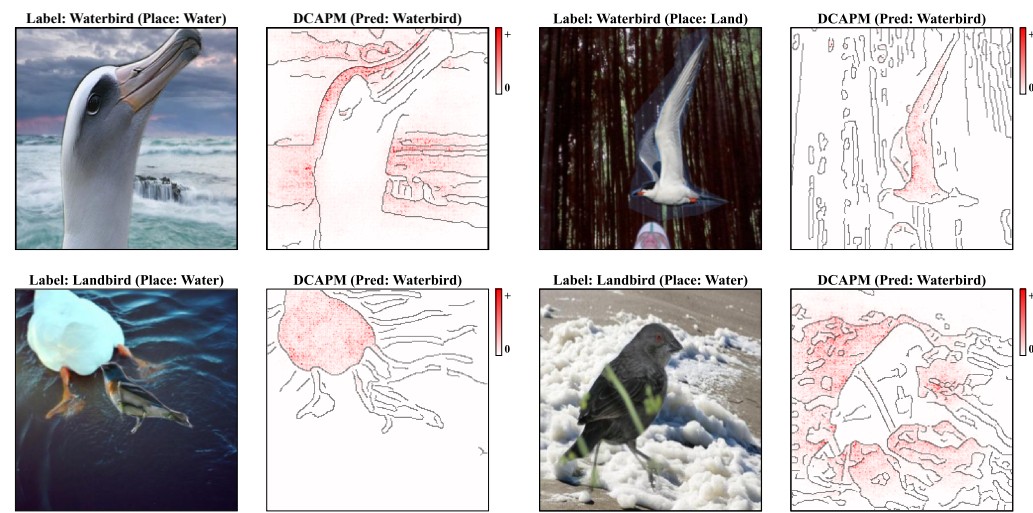

Figure 8: **Spurious correlation analysis via attribution maps.** (Top Left) **Aligned Correct:** The model highlights both the waterbird and the water, with a stronger focus on the water background, indicating a reliance on spurious correlations (lazy prediction). (Top Right) **Conflict Correct:** The model successfully overrides bias, exclusively highlighting the waterbird object without background noise. (Bottom) **Conflict Incorrect:** The model succumbs to bias by focusing on background features rather than the target object, such as a duck as a background object (Left) or the sea texture (Right), leading to misclassification.

**Experimental Setup**    In this experiment, we conducted a spurious correlation analysis using the Waterbirds dataset [3]. This dataset is designed for a binary classification task to distinguish between Waterbirds and Landbirds. Each image contains background (Place) information, consisting of either Water or Land. Consequently, the entire dataset comprises four groups: Landbird on Land, Landbird on Water, Waterbird on Land, and Waterbird on Water. For model training, we utilized the ResNet50 architecture (He et al., 2016) and followed the fine-tuning protocol provided in the group_DRO (Sagawa et al., 2020) [4]. The final test accuracy of the trained model is 77.22%.

**Hypothesis**    We employ attribution methods to distinguish whether the model relies on spurious (place) or core (bird) features. Our hypotheses are: **(1) Aligned Group and Correct Prediciton (Lazy Prediction):** The model likely exploits the background as an easy shortcut rather than focusing on the object. **(2) Conflict Group and Correct Prediction (Bias Override):** To predict correctly

---

[3]`https://huggingface.co/datasets/grodino/waterbirds`
[4]`https://github.com/kohpangwei/group_DRO/tree/master`

against a conflicting background, the model suppresses spurious signals and focuses on the bird. **(3) Conflict Group and Incorrect Prediction (Succumb to Bias):** Misclassification results from the model focusing on the misleading background, confirming reliance on spurious correlations.

**Qualitative Analysis Results**  To verify these hypotheses, we extracted and analyzed attribution maps for each case. As shown in Figure 8, the experimental results align with our hypotheses, and DCAPM successfully filters out noisy dummy features while clearly highlighting the spurious correlations learned by the model, demonstrating its utility for precise model debugging. In the **Conflict Correct** case (Top Left), we observe that the influence of the background (Land) is minimized, while the shape of the Waterbird object is clearly highlighted. Conversely, in the **Conflict Incorrect** cases (Bottom), high attribution values are distributed across the background region, visually demonstrating that the model was misled by background information, resulting in misclassification. Finally, in the **Aligned Correct** case (Top Right), significant attribution occurs not only on the object but also on the background. This confirms that the model utilizes background information as a primary basis for prediction (spurious correlation) during the training process.

**Additional Spurious Correlation Analysis Results**  Additionally, spurious correlations can be seen in the imagenet dataset. As shown in Figure 9, DCAPM not only successfully eliminates noisy attributions caused by dummy features and spurious Shapley interactions but also clearly highlights the spurious correlation features relevant to the decision, proving its capability to distinguish actual learned biases from mathematical noise.

| **Image** | **DCAPM** | **IG** |

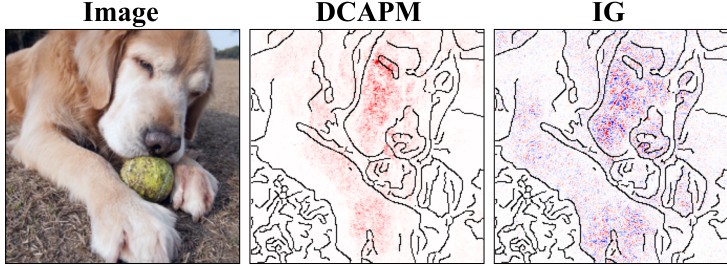

Figure 9: **Spurious correlation analysis:** DCAPM reveals the spurious correlation of the ImageNet2012 training dataset, which frequently associates a 'Golden Retriever' image with tennis balls.

## C.6 ENTROPY ANALYSIS ON IMAGE DOMAIN

In this section, we evaluate the sparsity of attribution maps by analyzing the entropy distribution across 500 randomly sampled images from the ImageNet validation set. As illustrated in Figure 10, all baseline methods exhibit relatively high entropy distributions, clustering around means ranging from 14.71 to 15.52. In contrast, DCAPM achieves a significantly lower mean entropy of 12.46. This distinct distribution shift confirms that our method yields the most sparse explanations, effectively concentrating attribution on relevant features while nullifying irrelevant noise.

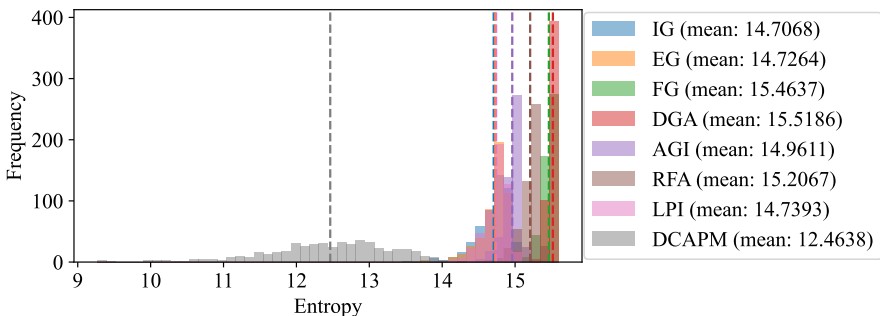

Figure 10: **Entropy analysis:** comparing baselines and DCAPM using VGG16 on 500 random samples from ImageNet2012 validation set.

## C.7 QUANTITATIVE EVALUATION ON THE FLOWERS AND OXFORD-PET DATASETS

To further validate the generalizability of our proposed method, we conducted a quantitative evaluation on the Flowers and Oxford-Pet datasets using a fine-tuned VGG16 model (see Appendix D for configuration details).

The results, summarized in Table 6, demonstrate that our method achieves superior performance in terms of robustness and reliability. Specifically, our approach obtains the best (lowest) scores for both Non-Sensitivity and Consistency across both datasets, significantly outperforming all baselines. For instance, on the Flowers dataset, our method achieves a Consistency score of 0.00027, an order of magnitude improvement over the next-best method, RFA (0.00232). While other methods such as FG and DGA excel on the DiffID metric, our approach remains highly competitive. These findings confirm that our method consistently generates more stable and faithful explanations across diverse data domains.

Table 6: **Quantitative evaluation of sensitivity and consistency on the Flowers and Oxford-Pet datasets.** Scores were computed using a fine-tuned VGG16 model on 500 randomly sampled instances from the validation set (fixed seed of 42). Higher is better for DiffID (↑), while lower is better for Non-Sensitivity (↓) and Consistency (↓). The **first** and **second**-best scores in each column are highlighted.

| | DiffID (↑) | | Non-Sensitivity (↓) | | Dummy Consistency (↓) | |
|---|---|---|---|---|---|---|
| | Flowers | OxfordPet | Flowers | OxfordPet | Flowers | OxfordPet |
| IG | $0.014 \pm 0.012$ | $0.004 \pm 0.003$ | $0.00168 \pm 0.00$ | $0.00158 \pm 0.00$ | $0.20642 \pm 0.05$ | $0.19507 \pm 0.05$ |
| EG | $0.022 \pm 0.019$ | $0.005 \pm 0.004$ | $0.00167 \pm 0.00$ | $0.00160 \pm 0.00$ | $0.13257 \pm 0.05$ | $0.08331 \pm 0.05$ |
| Saliency | $0.011 \pm 0.009$ | $0.003 \pm 0.002$ | $0.00167 \pm 0.00$ | $0.00160 \pm 0.00$ | $0.08691 \pm 0.05$ | $0.04385 \pm 0.04$ |
| I×G | $0.018 \pm 0.019$ | $0.004 \pm 0.003$ | $0.00170 \pm 0.00$ | $0.00163 \pm 0.00$ | $0.07141 \pm 0.06$ | $0.02281 \pm 0.03$ |
| FG | $\mathbf{0.061 \pm 0.046}$ | $\mathbf{0.027 \pm 0.022}$ | $0.00131 \pm 0.00$ | $0.00094 \pm 0.00$ | $0.00913 \pm 0.01$ | $0.00430 \pm 0.00$ |
| DGA | $\mathbf{0.061 \pm 0.044}$ | $\mathbf{0.028 \pm 0.024}$ | $0.00138 \pm 0.00$ | $\mathbf{0.00101 \pm 0.00}$ | $0.07214 \pm 0.03$ | $0.03326 \pm 0.02$ |
| AGI | $0.023 \pm 0.025$ | $0.006 \pm 0.005$ | $0.00171 \pm 0.00$ | $0.00154 \pm 0.00$ | $0.17358 \pm 0.06$ | $0.16858 \pm 0.06$ |
| RFA | $0.044 \pm 0.039$ | $0.023 \pm 0.021$ | $0.00101 \pm 0.00$ | $\mathbf{0.00071 \pm 0.00}$ | $\mathbf{0.00232 \pm 0.00}$ | $\mathbf{0.00196 \pm 0.00}$ |
| **DCAPM** | $\mathbf{0.067 \pm 0.044}$ | $0.015 \pm 0.022$ | $\mathbf{0.00018 \pm 0.00}$ | $\mathbf{0.00017 \pm 0.00}$ | $\mathbf{0.00027 \pm 0.00}$ | $\mathbf{0.00031 \pm 0.00}$ |

## C.8 OPTIMIZATION PROGRESS VISUALIZATION

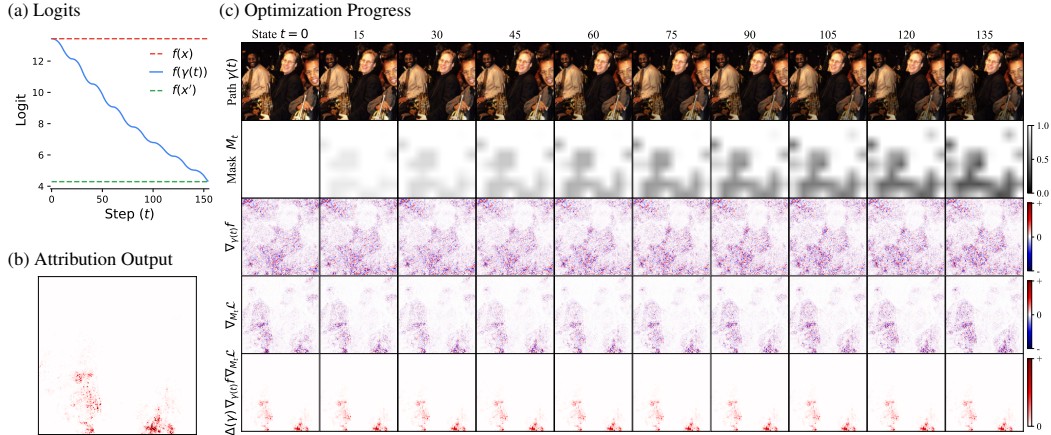

Figure 11: **The optimization process of target class 'French Horn'.** As the mask is optimized, the dummy players corresponding to the target class approach 1, while the remaining players approach 0, leading the function value to converge toward $f(\mathbf{x}')$. The target class is the predicted class at the first iteration.

To empirically verify our claim regarding *shattered gradients* and *rare zero gradients* in piecewise-linear networks, we conducted an experiment to analyze the gradient characteristics of a standard ReLU-based DNN.

**Experimental Setup**   We used a pretrained VGG16 model and the COCO 2017 dataset (Lin et al., 2014), leveraging its provided object segmentation masks to isolate background pixels. Our experiment is based on the premise that for a given target object classification, pixels in the image background can serve as a practical proxy for irrelevant or *dummy* features. We then computed the gradients of the model's classification output with respect to these background pixels. Our analysis focuses on two key aspects: (1) the distribution of these gradient values and (2) the sparsity of gradients across different layers of the network.

**Empirical Verification of "*Rare Zero and Noisy Near-Zero Gradients*"**   Our premise was that the gradients in irrelevant background regions would not be exactly zero but would instead form a noisy, near-zero distribution. The statistics gathered from different layers of the VGG16 model, presented in Table 7, strongly support this claim.

As shown in Table 7, the mean of the background gradient distribution across all layers is extremely close to zero. However, the standard deviation is consistently non-zero. This empirically demonstrates that identifying irrelevant features by simply checking for 'gradient == 0' is infeasible in practice. Instead, the gradients form a *noisy near-zero* distribution, justifying the need for methods robust to such noise.

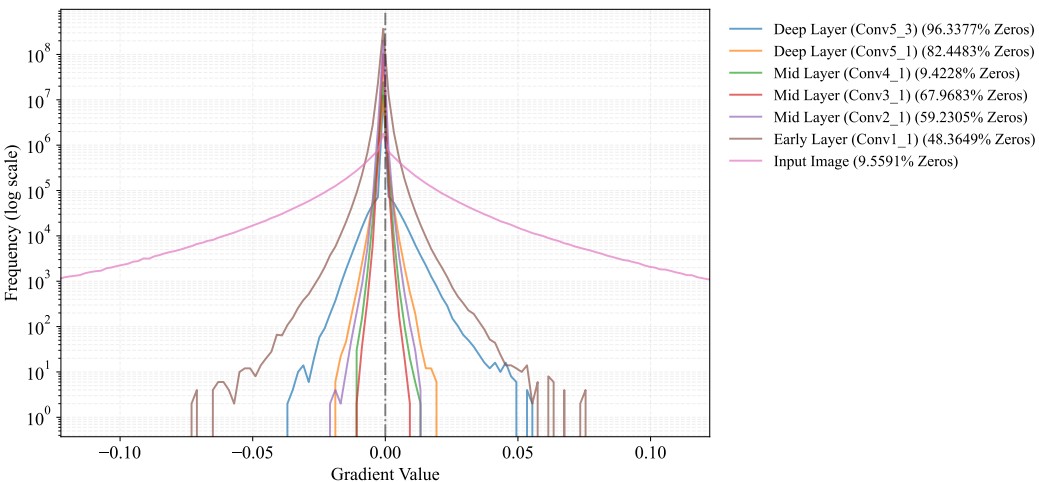

Figure 12: **Distribution of masked gradients corresponding to background pixels in the input image**, analyzed at different layers within a VGG16 model.

Table 7: **Statistical properties of background gradients across different VGG16 layers.**

| Layer | Entropy | Mean (approx.) | Std. Dev. (approx.) | Min | Max |
|---|---|---|---|---|---|
| Deep (Conv5_3) | 0.365 | $2.88 \times 10^{-5}$ | 0.00104 | -0.036 | 0.057 |
| Deep (Conv5_1) | 1.194 | $1.28 \times 10^{-5}$ | 0.00063 | -0.017 | 0.020 |
| Mid (Conv4_1) | **2.680** | $-7.46 \times 10^{-6}$ | 0.00045 | -0.010 | 0.015 |
| Mid (Conv3_1) | 1.233 | $1.21 \times 10^{-6}$ | 0.00017 | -0.009 | 0.009 |
| Mid (Conv2_1) | 1.172 | $7.94 \times 10^{-7}$ | 0.00027 | -0.019 | 0.014 |
| Early (Conv1_1) | 0.878 | $5.18 \times 10^{-7}$ | 0.00087 | -0.073 | 0.077 |
| **Input Image** | **1.826** | $\mathbf{2.70 \times 10^{-6}}$ | **0.02367** | **-1.051** | **0.957** |

**Empirical Verification of "*Shattered Gradients*" via Sparsity Analysis**     Our second premise was that gradients become less sparse and more *shattered* as they propagate backward to earlier layers (i.e., closer to the input). We verified this by measuring the percentage of exact zero gradients at different network depths, with results presented in Table 8.

The results show a clear, general trend: the percentage of zero gradients is very high in the deep layers near the output, which is expected due to the pruning nature of ReLU activations. However, this sparsity dramatically **decreases in earlier layers**, with the input layer gradients having only $\sim$9.6% exact zeros. This demonstrates that as the gradient signal propagates backward, it becomes increasingly dense and *shattered*, affecting nearly all input features, including the irrelevant ones in the background.

Table 8: **Sparsity of gradients across different VGG16 layers**, measured by the percentage of exact zeros.

| Layer | Location | Percentage of Exact Zeros |
|---|---|---|
| Deep (Conv5_3) | Close to Output | 96.34% |
| Deep (Conv5_1) | Close to Output | 82.45% |
| Mid (Conv4_1) | Middle of Network | 9.42% |
| Mid (Conv3_1) | Middle of Network | 67.97% |
| Mid (Conv2_1) | Middle of Network | 59.23% |
| Early (Conv1_1) | Close to Input | 48.36% |
| **Input Image** | Input | **9.56%** |

In conclusion, these empirical results validate our claims regarding the nature of gradients in piecewise-linear networks. They provide a strong motivation for our work, which aims to develop a method that can reliably handle both *noisy near-zero* and *shattered* gradients.

## C.10   EMPIRICAL VALIDATION OF SPURIOUS "*Dummy Interactions*"

To empirically validate our claim that feature interactions are captured when an attribution path traverses different linear regions, and that this can lead to non-zero attribution for dummy features, we conducted the following experiment.

**Experimental Setup**     We trained a three-layer Multi-Layer Perceptron (MLP) with a (4, 4, 2) architecture and two ReLU activation layers. The model was trained for binary classification on the TwoMoons synthetic dataset, on which it achieved 100% test accuracy. We analyzed the behavior of the gradient field along a straight-line path from an input point of $[-1.0, 2.4, 0.2]$ to a baseline of $[0, -0.5, 0]$. The input vector was augmented with a third dimension, $x_3$, which serves as a dummy feature independent of the classification task. Inspired by the approximation method for the Shapley interaction value from Chang et al. (2025), we then measured the change in the partial derivatives, $\Delta(\nabla f)$, between adjacent discrete steps ($t$ and $t-1$) along this path. The detailed method is as followed:

**Methodology**     Let $f : \mathbb{R}^{N=3} \to \mathbb{R}$ be a differentiable function, $\mathbf{x} \in \mathbb{R}^{N=3}$ be an input, and $\mathbf{x}' \in \mathbb{R}^{N=3}$ be a baseline. We define a straight-line path between them, discretized into $T$ steps, where a point on the path at step $t \in \{0, \ldots, T\}$ is given by $\mathbf{x}^t = \mathbf{x}' + \frac{t}{T}(\mathbf{x} - \mathbf{x}')$.

Inspired by the difference-of-gradients approach to approximating the Shapley interaction value from Chang et al. (2025), we define a metric to isolate the interaction between two features, $i$ and $j$, during a single path step from $t-1$ to $t$. Unlike prior work that perturbs a feature to its baseline value, we measure the effect of the infinitesimal movement of feature $j$ on the gradient of feature $i$.

To achieve this, we define a counterfactual point $\bar{\mathbf{x}}^{t-1}$ which is identical to $\mathbf{x}^{t-1}$ except that feature $j$ has moved to its position at step $t$:

$$\bar{\mathbf{x}}^{t-1} := (x_1^{t-1}, \ldots, x_{j-1}^{t-1}, x_j^t, x_{j+1}^{t-1}, \ldots, x_n^{t-1})$$

The interaction contribution of $j$ on $i$ during the step from $t-1$ to $t$, denoted as $I_{i,j}^{(t)}$, is then defined as the change in the partial derivative of $f$ with respect to $x_i$ between the original point $\mathbf{x}^{t-1}$ and the counterfactual point $\bar{\mathbf{x}}^{t-1}$, scaled by the displacement of $x_i$:

$$I_{i,j}^{(t)} := \left( \left.\frac{\partial f}{\partial x_i}\right|_{\mathbf{x}=\mathbf{x}^{t-1}} - \left.\frac{\partial f}{\partial x_i}\right|_{\mathbf{x}=\bar{\mathbf{x}}^{t-1}} \right) \cdot (x_i^{t-1} - x_i^t)$$

Here, the term within the parenthesis isolates the change in the gradient of feature $i$ caused solely by the movement of feature $j$ in that step. Summing $I_{i,j}^{(t)}$ over all steps $t = 1, \ldots, T$ would yield the total interaction between $i$ and $j$ along the path.

In this experiment, we omit the displacement term $(x_i^{t-1} - x_i^t)$. Because we use a straight-line path, this term simplifies to a constant scaling factor for each feature, which can be disregarded for the purpose of identifying non-zero interactions.

**Results**  The change in partial derivatives was zero for most steps, indicating the path remained within a single linear region. However, at specific intervals where the path crossed linear region boundaries, we observed significant, non-zero changes. The results are denoted as $\{t : [\Delta(\partial f/\partial x_1), \Delta(\partial f/\partial x_2), \Delta(\partial f/\partial x_3)]\}$:

- Step 5: $[5.6, -1.2, -\mathbf{0.0}]$
- Step 7: $[0.1, 0.5, \mathbf{0.1}]$

Critically, at Step 7, the traversal of a linear region boundary resulted in a non-zero change in the partial derivative with respect to the dummy feature ($x_3$). This empirically demonstrates how spurious interactions, arising from the piecewise-linear structure of the network, can be assigned to irrelevant features. A path integral method, by aggregating these local changes, would consequently assign a non-zero attribution to the dummy feature. This experiment validates our claim that path-dependence in rectifier networks is a primary cause of axiomatic violations and motivates the need for a path-finding method robust to these dummy interactions.

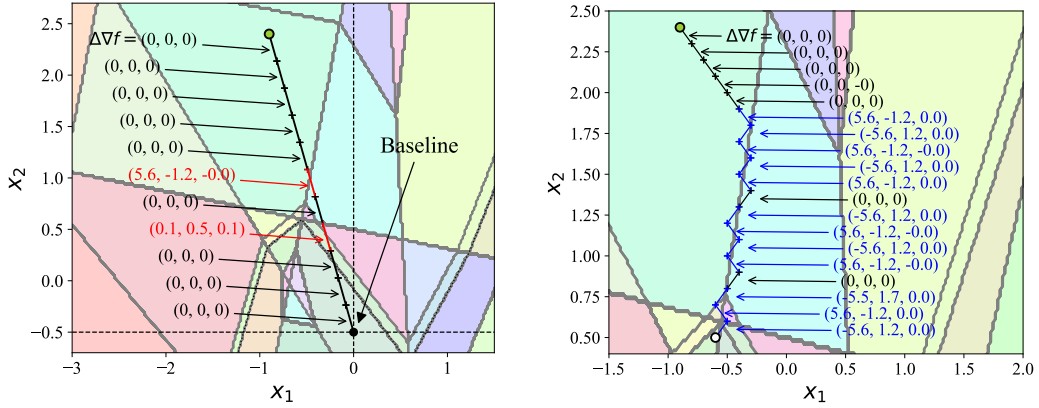

Figure 13: **Demonstration of a dummy interaction caused by traversing linear region boundaries.** A straight-line path (left) crossing the model's decision boundaries (background heatmap) induces a spurious, non-zero interaction effect on the irrelevant dummy feature $x_3$. In contrast, DCAPM (right) finds a path that effectively reduces the spurious feature interaction on $x_3$.

## C.11  VALIDATION OF THE PROPOSED DUMMY CONSISTENCY METRIC

We validate our proposed metric through theoretical grounding and quantitative analysis. Theoretically, the metric is a direct operationalization of the Dummy Consistency Axiom. Quantitatively, we demonstrate that our metric captures a novel dimension of attribution quality, distinct from existing sensitivity measures. To do this, we performed a correlation analysis using the attribution results from ten methods on 500 ImageNet samples with a VGG16 model. As shown in Table 9, we report the Pearson and Spearman correlations between our Dummy Consistency metric and four established

sensitivity metrics: Insertion/Deletion Games (Petsiuk et al., 2018), DiffID (Yang et al., 2023), and Non-Sensitivity (Nguyen & Martínez, 2020). The results show a statistically significant but low positive correlation between our metric and the others. This finding is critical: it empirically confirms that Dummy Consistency is a distinct and complementary property, not a redundant one. Therefore, improving consistency does not require a trade-off with sensitivity; rather, it contributes to a more holistically robust attribution.

Table 9: **Correlation between our Dummy Consistency metric and four established sensitivity metrics.** Pearson and Spearman coefficients are reported, with **p-values** in parentheses. Our metric is shown to be largely orthogonal to existing sensitivity measures. '1-DConsistency' denotes the value of '1-(normalized Dummy Consistency score)', used to align the interpretation direction (i.e., higher is better). (***: p-value < 0.001).

| | Insertion (↑) | 1-Deletion (↑) | DiffID (↑) | 1-Nonsensitivity (↑) |
|---|---|---|---|---|
| | **Pearson Correlation** | | | |
| | 0.226 (***) | 0.093 (***) | 0.167 (***) | 0.049 (***) |
| 1-DConsistency (↑) | **Spearman Rank Correlation** | | | |
| | 0.273 (***) | 0.107 (***) | 0.322 (***) | 0.019 (***) |

### C.12 NOISE ROBUSTNESS TEST

Single-instance attribution methods such as Saliency Map (Simonyan et al., 2013) and FG (Srinivas & Fleuret, 2019) can be sensitive to input noise, potentially undermining the reliability of their explanations. As a path-based method, our approach is designed to offer greater robustness against such perturbations.

Figure 14 presents a qualitative and quantitative evaluation of this robustness. We compare our method against IG and FG on a single 'birdhouse' sample, to which we incrementally add Gaussian noise with increasing standard deviation ($\sigma$ from 0 to 1). On the DiffID sensitivity metric, our method demonstrates robustness comparable to IG, maintaining competitive performance even at high noise levels. The primary advantage of our approach, however, is observed in the dummy consistency metric. While IG and FG produce high MSE scores due to spurious attributions on the noisy background pixels, our method maintains a substantially lower MSE, highlighting its ability to consistently focus on true signal features.

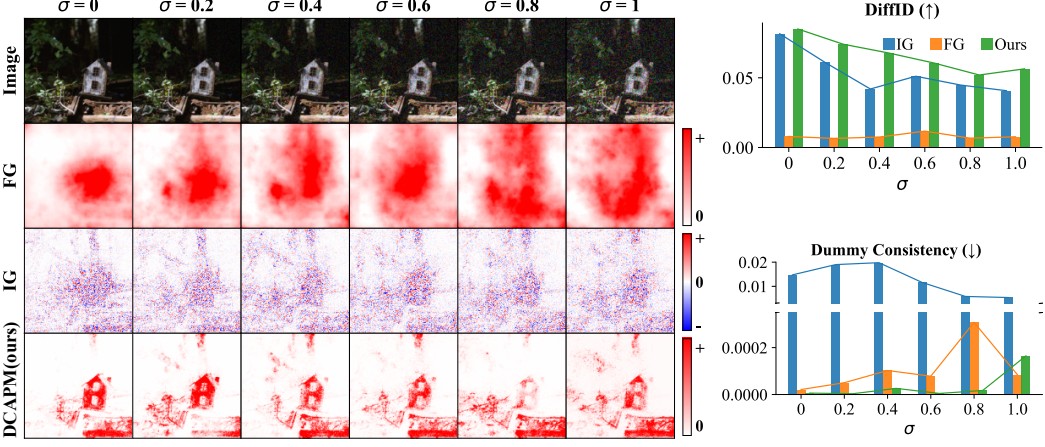

Figure 14: **Noise robustness evaluation for the 'Birdhouse' class, comparing our method against IG and FG.** (a) Visual comparison of attribution maps under increasing levels of additive noise. (b) DiffID scores (sensitivity), where higher is better. (c) Dummy Consistency scores (MSE), where lower is better.

# D  EXPERIMENTAL DETAILS

This section provides the experimental details for this study, including the benchmark methods, datasets, and hyperparameters employed across all experiments.

## D.1  DATASETS AND MODELS

### D.1.1  DATASETS

Our evaluation is conducted on four benchmark datasets: three for computer vision and one for natural language processing.

- **Flowers**[5]: A dataset of 4,317 object-centered images resized to $3 \times 224 \times 224$, containing five classes: *daisy*, *dandelion*, *rose*, *sunflower*, and *tulip*. The dataset was randomly split the data into training and validation sets using a 9:1 ratio, with a fixed random seed of 42 for reproducibility.
- **Oxford-IIIT Pet** (Parkhi et al., 2012): An object-centered dataset with 37 classes specifying species (*cat* or *dog*) and breed. It consists of 3,680 training and 3,669 testing images, all resized to $3 \times 224 \times 224$.
- **ImageNet-2012** (Deng et al., 2009): The full dataset comprising 1.28M training images and 50,000 testing images across 1,000 classes. All images were resized to $3 \times 224 \times 224$ and normalized using a mean of (0.485, 0.456, 0.406) and a standard deviation of (0.229, 0.224, 0.225).
- **IMDb** (Maas et al., 2011): A sentiment analysis dataset containing 50,000 movie reviews, evenly split into training and testing sets with balanced positive and negative labels.

Table 10: **Dataset and model summary.** An asterisk (*) denotes models that were fine-tuned and accuracy scores computed by us. Train/Test splits are indicated by a colon.

|  | **Flowers** | **Oxford-IIIT Pet** | **ImageNet-2012** | **IMDb** |
|---|---|---|---|---|
| Image Size | $3 \times 224 \times 224$ | $3 \times 224 \times 224$ | $3 \times 224 \times 224$ | N/A (Text) |
| # of Labels | 5 | 37 | 1,000 | 2 |
| Dataset Size | 4,317 | 3,680 : 3,669 | 1.28M : 50,000 | 25,000 : 25,000 |
| Model (hidden dim.) | VGG16* (1024) | VGG16* (1024) | VGG16 (4096) InceptionV3 (2048) ResNet18 (512) | BERT-Base (768) BERT-Large*(1024) |
| Accuracy | 0.9305* | 0.9021* | V: 0.7159, I: 0.6954, R: 0.6976 | Base: 0.8909 Large: 0.9464* |

### D.1.2  MODELS

We used the following pretrained models in our experiments:

- **Vision Models:** We employed standard **VGG16** (Simonyan & Zisserman, 2015), **InceptionV3** (Szegedy et al., 2016), and **ResNet18** (He et al., 2016) models. The classifier head dimensions were 4096 (VGG16), 2048 (InceptionV3), and 512 (ResNet18), respectively. For the Flowers and Oxford-IIIT Pet datasets, we fine-tuned VGG16 models with a reduced classifier dimension of 1024.
- **Transformer Models:** We evaluated two **BERT** (Devlin et al., 2019) models sourced from the `transformers` library. The first is **BERT-base** model (`textattack/bert-base-uncased-imdb`[6]), a 12-layer model with a 768-dimensional hidden state and 110 million parameters. The second is a fine-tuned **BERT-large** model (`jigarcpatel/fine-tuned-bert-large-uncased-IMDb-dataset`[7]), a 24-layer model with a 1024-dimensional hidden state and 336 million parameters. For the BERT-large experiments, the maximum sequence length was set to 256 tokens to ensure computational tractability.

---

[5]`https://www.kaggle.com/datasets/alxmamaev/flowers-recognition`
[6]`https://huggingface.co/textattack/bert-base-uncased-imdb`
[7]`https://huggingface.co/jigarcpatel/fine-tuned-bert-large-uncased-IMDb-dataset`

- **Synthetic Data Models:** For the 2D data experiments shown in Figures 1 and 13 (**Circles** and **TwoMoons** (Buitinck et al., 2013)), we trained a simple MLP with two ReLU-activated hidden layers, as detailed in Table 11. The models yielded test accuracies of 1.000 and 0.9045 for the Circles and TwoMoons datasets, respectively.

Table 11: **MLP architecture for the TwoMoons and Circles datasets.**

| Layer Type | Output Shape |
| --- | --- |
| Fully Connected + ReLU | 4 |
| Fully Connected + ReLU | 4 |
| Fully Connected | 2 |

## D.2 Benchmark Methods and Implementations

To evaluate our approach, we compare it against a comprehensive set of benchmark methods chosen for their relevance to attribution consistency in Piecewise Linear Neural Networks (PLNNs). We group these methods into the following categories: **Gradient-based Methods:** FullGrad (FG) (Srinivas & Fleuret, 2019), Distilled Gradient Aggregation (DGA) (Jeon et al., 2022), Saliency Maps (Simonyan et al., 2013), and Gradient×Input(G*I) (Shrikumar et al., 2017). **Path and Integration-based Methods:** Adversarial Gradient Integration (AGI) (Pan et al., 2021), Recalibrating Feature Attributions (RFA) (Yang et al., 2022), and Local Path Integration (LPI) (Yang et al., 2023). **Axiomatic Methods:** Integrated Gradients (IG) (Sundararajan et al., 2017) and Expected Gradients (EG). For all experiments, we used the official open-source implementations of each method with their default hyperparameter settings.

- **Integrated Gradients (IG)** (Sundararajan et al., 2017) is an axiomatic feature attribution technique grounded in Aumann-Shapley values. It approximates the Shapley value by integrating gradients along a straight-line path from a single baseline input to the original input.

- **Expected Gradients (EG)** (Erion et al., 2021) builds upon IG by reformulating the path integral as an expectation over multiple reference values sampled from a background distribution. This converges to attributions that sum to the difference between the model's expected output and its current output for a given input.

- **FullGrad (FG)** (Srinivas & Fleuret, 2019) avoids path integration to mitigate counterintuitive attribution issue that can arise when paths cross linear region boundaries. It combines local input gradients with bias gradients from throughout the network, which are heuristically upsampled to the input dimension. By doing so, it sacrifices the efficiency axiom but compensates with bias terms. FG introduces the *weak dependence* property, which, unlike *input invariance*, allows two distinct inputs to have different attributions even if they reside in the same, potentially disconnected, linear region.

- **Distilled Gradient Aggregation (DGA)** (Jeon et al., 2022) is a variant of FG that iteratively refines attributions through a masking process. It employs two masks: one to filter out weakly contributing features (noise) and another to regularize features with excessively high contributions. The final attribution is an aggregation of these masked features, using FG as the base explainer.

- **Adversarial Gradient Integration (AGI)** (Pan et al., 2021) is a path-based method that finds both a path and multiple baselines adversarially. Instead of a straight line, AGI defines a non-linear path by following the gradient ascent direction on the model's loss surface until the prediction for the target class is nullified. This process is stabilized using techniques from adversarial attacks, such as Projected Gradient Descent (PGD).

- **Recalibrating Feature Attributions (RFA)** (Yang et al., 2022) extends AGI by defining criteria for valid path interpolations for aggregating positive attributions. To achieve this, RFA posits that a valid baseline should represent the absence of the input signal, and the path from such a baseline should follow the gradient ascent direction in the model's logit surface. This validity condition is applied on a feature-wise basis across multiple baselines sampled from the training data.

- **Local Path Integration (LPI)** (Yang et al., 2023) also aggregates gradients over multiple baselines, but constrains the selection of these baselines to the same linear region as the input to consider the

property of piecewise linear neural networks (Chu et al., 2018). However, its method for estimating this region—K-Means clustering in the output logit space—is an indirect heuristic. Identifying points that share the exact same linear function is practically infeasible due to the extremely high number of tiny linear regions (polytopes) in DNNs.

### D.3 HYPERPARAMETERS FOR ATTRIBUTION COMPUTATION

The hyperparameters for the attribution methods used in our experiments were set as follows.

**Vision Models (VGG16, ResNet18, InceptionV3)**   For vision models, the dummy-mask optimization process, applied in both the attribution and evaluation phases of our method, was configured as follows. We set the mask resolution to $h \times w = 8 \times 8$ and employed a triangular cyclical learning rate policy (Smith, 2017) with a base rate of 0.0001, a maximum rate of 0.01, and a cycle step size of 10. The L1 regularization factor ($\lambda$) was set to 0.0001, and the threshold for identifying dummy interactions ($\tau$) was 0.1. The optimization was run for a maximum of 600 steps, using a constant $\epsilon$ of $1 \times 10^{-4}$ within a termination condition for numerical stability while satisfying the $\epsilon$-monotonicity axiom.

**Language Models (BERT-Base, BERT-Large)**   For experiments involving BERT models, we used a distinct set of hyperparameters for our method (DCAPM). The mask size was adapted to match the model's hidden dimension: 768 for BERT-base and 1024 for BERT-large. The optimization was configured with a learning rate of 0.001, an L1 regularization factor ($\lambda$) of 0.001, and a dummy interaction threshold ($\tau$) of 0.1, running for 600 iterations with $\epsilon = 1 \times 10^{-4}$. For these language model experiments, both DCAPM and the IG benchmark used a zero-vector baseline.

### D.4 EXPERIMENTAL DETAILS FOR FIGURE 1

This experiment is designed to demonstrate the vulnerability of standard attribution methods to the introduction of a single, semantically irrelevant feature on the rectified DNNs. We show that both path-based methods (e.g., IG) and single-instance gradient-based methods (e.g., FG) produce distorted attributions, violating core axioms, Dummy Player and Dummy Consistency axioms, when presented with such an input.

**Experimental Setup**   The experimental setup reproduces the 2D Circles classification task from (Jeon et al., 2022, Figure 4). We trained a three-layer Multi-Layer Perceptron (MLP) with two ReLU activation layers, which achieved 100% test accuracy on this task. To the original 2D input space, we introduce an irrelevant dummy feature axis ($x_3$). Values for this dummy feature were sampled from a uniform distribution over the range of the original dataset features.

**Key Observation**   The central finding, illustrated in Figure 1, is that the introduction of this single dummy feature causes both IG and FG to produce distorted and counter-intuitive attributions for the true signal features ($x_1, x_2$). This demonstrates a clear violation of the Dummy Player and Dummy Consistency axioms and motivates the need for attribution methods that are robust to such axiomatic failures.

### D.5 NON-SENSITIVITY METRIC AND ITS MODIFICATION

The *Non-Sensitivity* metric, originally proposed by (Nguyen & Martínez, 2020), serves as a quantitative measure for the Dummy Player axiom, as it evaluates the alignment between a feature's attribution and its functional relevance to the model's output. In this work, we compare the Non-Sensitivity metric with our proposed metric. However, due to the unavailability of an official open-source implementation, we first reintroduce the original definition and then detail our own modified implementation.

**Definition 5** (Non-Sensitivity (Nguyen & Martínez, 2020))**.** *Let $A_0 = \big\{\{i \in \{1, \ldots, N\} \mid a_i = 0\}$ denote the set of feature indices assigned zero attribution. Further, let $X_0 = \big\{i \in \{1, \ldots, N\}\big\} \mid \mathbb{E}\big[\|f(x) - f(x_{\neg i})\|^2\big] = 0\big\}$. be the set of indices corresponding to features on which the model $f$ is*

*not functionally dependent. Then, the Non-Sensitivity metric is defined as:*

$$\text{NonSens}(a, f) = |A_0 \triangle X_0| \tag{12}$$

*where $|\cdot|$ denotes the cardinality of a set, and $\triangle$ represents the symmetric difference (XOR) between the two sets.*

The original metric penalizes an attribution method if a feature with zero attribution ($i \in A_0$) is actually functionally relevant ($i \notin X_0$), or vice-versa. However, this formulation presents practical challenges, as exact zero attributions and perfectly invariant outputs are rare in real-world DNNs. Furthermore, the raw score is not normalized, making comparisons across different feature spaces difficult. To address these limitations, we propose a more practical, normalized version of the Non-Sensitivity metric that incorporates a tolerance threshold and operates on feature partitions.

**Definition 6** (Modified Non-Sensitivity). *Let $a_i$ denote the attribution scores sorted in ascending order by their absolute values, and $x_i$ be the corresponding input features. We define a uniform partition $\mathcal{P} = \{S_k\}_{k=1}^K$, where each subset $S_k = \{x_{(\mathcal{P}_k)}, \ldots, x_{(\mathcal{P}_{k+1})}\} \subset N$ contains a consecutive segment of features from the ordered list. Given a tolerance threshold $\epsilon > 0$, we define dummy players with respect to the attribution and the model's functional behavior as follows:*

$$A_0 = \{S_k \subset N \mid \exists i \in S_k, \|a_i\| < \epsilon\}, \tag{13}$$

$$X_0 = \{S_k \subset N \mid \|f(x) - f(x_{\neg S_k})\| < \epsilon\}, \tag{14}$$

*where $f(x_{\neg S_k})$ denotes the function output when features in $S_k$ are replaced by their global reference point values, keeping the remaining features unchanged. The modified Non-Sensitivity metric is then defined as:*

$$\text{Modified Non-Sensitivity} = \frac{|A_0 \triangle X_0|}{|\mathcal{P}|}, \tag{15}$$

*The metric is normalized by $K = |\mathcal{P}|$, the total number of partitions. A lower score on this metric signifies a better alignment between the attribution scores and the model's functional dependencies.*

# E  ABLATION STUDY

## E.1  ABLATION STUDY: THE IMPACT OF NULLIFYING DUMMY INTERACTIONS

Our proposed method consists of two key optimization phases: (1) dummy feature selection and (2) dummy interaction nullification. To isolate the contribution of each phase, we perform an ablation study on VGG16 using 500 ImageNet samples. We first evaluate performance using only dummy feature selection and then measure the additional gains from incorporating the interaction nullification term. The results in Table 12 demonstrate that while feature selection alone provides a strong baseline, the explicit nullification of dummy interactions is crucial for achieving the best performance across both sensitivity and dummy consistency metrics.

Table 12: **Ablation study evaluating the impact of nullifying dummy interactions.** The metrics of Sensitivity and Dummy Consistency are reported on VGG16 using 500 samples from the ImageNet validation set.

| Phase Configuration | DiffID($\uparrow$) | NonSens($\downarrow$) | DumCons($\downarrow$) |
|---|---|---|---|
| (1) Dummy Feature Selection Only | .0334 $\pm$.05 | .00190 $\pm$.00 | .005985 $\pm$.03 |
| **(1) + (2) With Dummy Interaction Nullification** | **.0362** $\pm$.06 | **.00023** $\pm$.00 | **.000398** $\pm$.00 |

## E.2  ABLATION STUDY ON HYPERPARAMETERS

The proposed method involves three key hyperparameters: the L1 regularization scaling factor, $\lambda$, the threshold for identifying dummy interactions, $\tau$, and mask size $h, w$ for the optimization parameter. We conducted ablation studies to analyze their impact on performance.

**Analysis of Regularization Factor** $\lambda$   Ablation results for $\lambda$ are presented in Table 13. The results indicate that $\lambda = 0.0001$ provides the best trade-off, achieving optimal sensitivity scores (DiffID and NonSens) without significantly compromising the dummy consistency score. While the regularization term plays a role in fine-tuning performance, its overall impact is modest. This suggests that the effectiveness of our method is primarily driven by its structural design—the dummy feature selection-based path-finding—rather than by hyperparameter optimization. Based on this, we selected $\lambda = 0.0001$ for all experiments in this paper.

Table 13: **Ablation study on the scaling factor $\lambda$ of the regularization term.** Sensitivity and Consistency scores are reported for our method with varying values of $\lambda$, evaluated using the VGG16 model and 500 samples from the ImageNet validation set.

| $\lambda$ | DiffID($\uparrow$) | Non-Sensitivity($\downarrow$) | Dummy Consistency($\downarrow$) |
|---|---|---|---|
| 0 | .0299 $_{\pm.05}$ | .00056 $_{\pm.00}$ | .000472 $_{\pm.00}$ |
| 0.1 | .0334 $_{\pm.05}$ | .00190 $_{\pm.00}$ | **.000385** $_{\pm.00}$ |
| **0.0001** | **.0362** $_{\pm.06}$ | **.00023** $_{\pm.00}$ | .000398 $_{\pm.00}$ |

**Analysis of Dummy Interaction Threshold** $\tau_g$   For the threshold $\tau_g$, the results in Table 14 show that our method is largely robust to its value. The performance metrics exhibit minimal variation across the tested range, confirming the stability of our approach with respect to this parameter. Based on this finding, we selected $\tau_g = 0.1$ for all main experiments.

Table 14: **Ablation study on the threshold $\tau_g$ used to identify dummy interactions based on gradient magnitudes.** Sensitivity and Consistency scores are reported for varying values of $\tau_g$, evaluated using the VGG16 model and 500 samples from the ImageNet validation set.

| $\tau_g$ | DiffID($\uparrow$) | Non-Sensitivity($\downarrow$) | Dummy Consistency($\downarrow$) |
|---|---|---|---|
| 0.001 | .0359 $_{\pm.06}$ | .00022 $_{\pm.00}$ | .000474 $_{\pm.00}$ |
| 0.01 | .0359 $_{\pm.06}$ | .00022 $_{\pm.00}$ | .000474 $_{\pm.00}$ |
| **0.1** | **.0362** $_{\pm.06}$ | .00023 $_{\pm.00}$ | **.000398** $_{\pm.00}$ |
| 0.2 | .0348 $_{\pm.06}$ | **.00019** $_{\pm.00}$ | .000465 $_{\pm.00}$ |

**Ablation Study on Mask Size**   We conducted an ablation study to evaluate the impact of mask size ($h = w \in \{4, 8, 16, 32\}$), holding all other hyperparameters constant as specified in Appendix D.3. The results, presented in Table 15, indicate that mask size is a sensitive hyperparameter, revealing a trade-off between the evaluation metrics. While larger mask sizes (16 and 32) yield optimal scores on individual metrics (DiffID and Dummy Consistency, respectively), a mask size of 8 provides the most effective balance. Specifically, the 8x8 mask achieves a substantially better Non-Sensitivity score than other configurations while maintaining highly competitive performance on the DiffID metric. Therefore, based on this trade-off, we selected a mask size of 8x8 for all main experiments.

Table 15: **Ablation study evaluating the impact of the mask size.** The metrics of Sensitivity and Consistency are reported on VGG16 using 500 samples from the ImageNet validation set.

| Mask Size | DiffID($\uparrow$) | Non-Sensitivity($\downarrow$) | Dummy Consistency($\downarrow$) |
|---|---|---|---|
| 4 | .03397 $_{\pm.05}$ | .00190 $_{\pm.00}$ | .00025 $_{\pm.00}$ |
| **8** | .03619 $_{\pm.06}$ | **.00023** $_{\pm.00}$ | .00040 $_{\pm.00}$ |
| 16 | **.03647** $_{\pm.04}$ | .00036 $_{\pm.00}$ | .00019 $_{\pm.00}$ |
| 32 | .03381 $_{\pm.07}$ | .00046 $_{\pm.00}$ | **.00017** $_{\pm.00}$ |

### E.3   ABLATION STUDY ON THE CYCLICAL LEARNING RATE

We conducted an ablation study to evaluate the effect of employing a cyclical learning rate, a technique intended to help the optimization process avoid poor local minima. The results, presented in Table 16,

reveal a clear trade-off between our primary evaluation metrics. While the cyclical learning rate improved the Dummy Consistency score, this came at the cost of a decrease in the DiffID sensitivity score. Given that maintaining high sensitivity is a critical requirement for a reliable attribution method, we concluded that the standard learning rate policy offered a superior overall performance balance. Consequently, all main experiments presented in this paper were conducted without the cyclical learning rate policy.

Table 16: **Ablation study evaluating the impact of the Cyclical Learning Rate.** The metrics of sensitivity and dummy consistency are reported on the VGG16 using 500 samples from the ImageNet validation set.

| Phase Configuration | DiffID($\uparrow$) | Non-Sensitivity($\downarrow$) | Dummy Consistency($\downarrow$) |
|---|---|---|---|
| **Without Cyclic LR** | .0362 ±.06 | .00023 ±.00 | .000398 ±.00 |
| With Cyclic LR | .0349 ±.06 | .00023 ±.00 | .000218 ±.00 |

## F    IMPLEMENTATION CODE

Below is the Python implementation of the our proposed algorithm, consistent attribution path method(DCAPM), and consistency metric.

```python
import torch

class DCAPM(object):
    def __init__(self,
        model,              # (func) Model function to be explained
        preprocess,         # (func) Predefined function to approximate reference point
        im_size = 224,      # (int) Input image size
        m_size = 8,         # (int) Mask optimization parameter size
        max_iter = 600,     # (int) Maximum number of iteration
        base_lr = 0.0001,   # (float) Base learning rate(LR) for cyclical LR
        max_lr = 0.01,      # (float) Maximum learning rate for cyclical LR
        step_size = 10,     # (int) Step size for cyclical LR
        eps = 1e-4,         # (float) Infinitesimal value
        tau = 0.1,          # (float) Threshold for dummy localization
        lambda_reg = 1e-4,  # (float) Scaling factor for L1 regularization
        ):

        self.model, self.preprocess = model, preprocess
        self.im_size, self.m_size = im_size, m_size
        self.max_iter, self.eps, self.tau = max_iter, eps, tau
        self.base_lr, self.max_lr, self.step_size = base_le, max_lr, step_size

    def get_cyclic_lr(step, base_lr, max_lr, step_size):
        cycle = step // (2 * step_size)
        x = step % (2 * step_size)
        scale = 1.0 - abs(x - step_size) / step_size
        lr = base_lr + (max_lr - base_lr) * scale
        return torch.tensor(lr)

    def get_mask(self, image):
        prev_masked_image = image.clone()
        input_size = (self.im_size, self.im_size)
        grid = torch.ones((1, 1, self.m_size, self.m_size), requires_grad=True)
        baseline = self.preprocess(torch.zeros_like(image.detach().cpu()))
        masked_attrs = torch.zeros((1,3,self.im_size,self.im_size))

        scores, masks, images, masked_attrs, FLAG = [], [], [], [], False
        C = torch.ones_like(grid)
        for self.iteration in range(self.max_iter):
            lr = self.get_cyclic_lr(step=self.iteration, self.base_lr,
                self.max_lr, self.step_size)
```

```
1836            # Resize the mask parameter to the original image size.
1837            mask = torch.nn.functional.interpolate(grid, input_size, mode='bilinear',
1838                          align_corners=False)
1839            mask.data = torch.clamp(mask.data, 0, 1)
1840            masked_image = image * mask + baseline * (1 - mask)
1841
1842            # Obtain the logit value from the masked image.
1843            with torch.no_grad():
1844                if self.iteration == 0:
1845                    original_score = self.model(image)
                        target = torch.argmax(original_score, -1)
1846                masked_score = self.model(masked_image)[range(masked_image.shape[0]), target]
1847            # Terminate based on predefined conditions.
1848            if torch.allclose(mask.reshape(-1).detach(),
1849                          prev_mask.reshape(-1), rtol=self.eps, atol=self.eps) or
1850                    prev_score - masked_score.item() < eps or
1851                    self.iteration == self.max_iter - 1 or masked_score < 0:
                    break
1852
1853            reg_loss = lambda_reg * torch.norm(mask, 1)
1854            loss = masked_score - reg_loss
                loss.backward(retain_graph=True)
1855
1856            # Compute the difference of gradients
1857            with torch.no_grad():
1858                gradient = grid.grad
1859                threshold = torch.quantile(torch.abs(grad), self.tau, interpolation='lower')
1860                C_tmp = gradient > threshold
                    grid = grid - gradient.sign() * C_tmp * C * lr
1861                grid.data = torch.clamp(grid.data, 0, 1)
1862                grid.requires_grad = True
                    C = grad > threshold
1863
1864            # Compute the gradient of the loss function w.r.t. the mask.
1865            mask_grad = lr * torch.autograd.grad(
1866                          outputs=loss, inputs=mask, retain_graph=True)[0].detach()
1867            # Compute the gradient of the self.model w.r.t. the masked image.
1868            fn_grad = torch.autograd.grad(
1869                          outputs=masked_score,
1870                          inputs=masked_image,
1871                          retain_graph=True)[0].detach()
1872            delta = prev_masked_image - masked_image
1873            self.masked_attrs += fn_grad * mask_grad * delta
1874            prev_masked_image = masked_image.clone().detach()
1875            prev_mask = mask.clone().detach()
                prev_score = masked_score.item()
1876
1877        return mask
1878
1879    def get_attribution(self, image):
1880        _ = self.get_mask(image)
1881        # Normalize by the number of iterations.
1882        attribution = self.masked_attrs.detach() / (self.iteration + 1)
1883
1884        return attribution
1885
1886    def consistency(
1887        orig_attribution,   # (tensor) Original attribution map to be evaluated.
1888        image,              # (tensor) Original image to be evaluated.
1888        explainer,          # (func) Target explainer to be evaluated.
1889        minmax_scale,       # (func) Minmax scaling function.
            tau=0.1,            # (float) Threshold for dummy consistency.
```

```
):
    ''' Dummy players estimation-based dummy consistency measure. '''

    _, C, H, W = image.shape
    HW = H * W
    step = (HW - 1) // (n_steps - 1)

    # Obtain the approximated global reference point.
    substrate_fn = torch.zeros_like
    baseline = explainer.preprocess(substrate_fn(image))

    # Get sorted attribution indices
    if orig_attribution.dim() == 4 and orig_attribution.shape[1] > 1:
        orig_attribution = orig_attribution.mean(dim=1)
    min_, max_ = orig_attribution.min(), orig_attribution.max()
    orig_attribution = minmax_scale(orig_attribution, min_, max_)
    threshold = torch.quantile(orig_attribution, tau, interpolation='lower')
    dummy_mask = orig_attribution > threshold
    masked_image = image * dummy_mask + baseline * (1 - dummy_mask)

    # Compute reduced attribution
    reduced_attr = explainer.get_attribution(masked_image).mean(dim=1)
    reduced_attr = minmax_scale(reduced_attr, min_, max_)
    reduced_attr = reduced_attr.view(1, HW)

    # Compute conditioned attribution
    condition_attr = orig_attribution.view(1, HW)

    # Calculate MSE only on the dummy cases (abs attribution < tau).
    mask = torch.abs(condition_attr) < threshold
    valid_count = mask.sum(dim=-1)
    mse = torch.pow(condition_attr * mask - reduced_attr * mask, 2).sum(dim=-1) /
                torch.where(valid_count > 0, valid_count, 1)
    mse = torch.nan_to_num(mse, nan=0.0)

    return mse.mean()
```

# G  COMPUTATIONAL EFFICIENCY AND SCALABILITY

## G.1  COMPUTATIONAL COMPLEXITY

In this section, we analyze the computational complexity of the proposed algorithm with respect to the input feature size, $N$. The time complexity is dominated by four primary operations performed at each of the $T_{\text{fin}}$ iterations: gradient computation, mask updates, attribution aggregation, and the nullification of dummy interactions. Since each of these operations is $\mathcal{O}(N)$, the total time complexity per sample is $\mathcal{O}(T_{\text{fin}} \cdot N)$, scaling linearly with the number of input features. The space complexity is also linear, requiring $\mathcal{O}(N)$ memory. This includes storage for the current and previous masks, the current and previous gradients, and the accumulated attribution values. As our proposed evaluation metric reuses the attribution computation process, its complexity is identical.

## G.2  COMPUTATIONAL RESOURCES AND RUNTIMES

All experiments were conducted on a single NVIDIA RTX A6000 GPU. We report the following empirical runtimes: For vision models, computing attribution for a single $224 \times 224 \times 3$ image required approximately **22.02 seconds** for 600 iterations. The total memory usage was approximately 2,270 MB per sample, which includes the classifier model parameters and intermediate gradients. For a language model BERT-Large model with a maximum sequence length of 512 tokens, the attribution computation required approximately **27 seconds** for 100 iterations on the same hardware.

# H MATHEMATICAL DEFINITIONS RELATED TO THE SHAPLEY VALUE

## H.1 THE ASSUMPTIONS OF AUMANN-SHAPLEY VALUE

Here, we provide the formal mathematical definitions for the key assumptions and axioms referenced in our theoretical framework.

**Definition 7** (Monotonicity (Non-decreasing)). *A function $f : \mathcal{X} \to \mathbb{R}$, where $\mathcal{X} \subseteq \mathbb{R}^{|N|}$, is **non-decreasing** if for any two input vectors $\mathbf{x}, \mathbf{y} \in \mathcal{X}$:*

$$\text{If } x_i \geq y_i \text{ for all } i \in N, \text{ then } f(\mathbf{x}) \geq f(\mathbf{y}).$$

**Definition 8** (Non-negativity of Function Output). *A function $f : \mathcal{X} \to \mathbb{R}$ has a **non-negative output** if its range is a subset of the non-negative real numbers, denoted $\mathbb{R}_+$:*

$$\forall \mathbf{x} \in \mathcal{X}, f(\mathbf{x}) \geq 0.$$

**Definition 9** (Differentiability and Continuity of Value Function). *A function $f : \mathcal{X} \to \mathbb{R}$ is **continuously differentiable** (denoted as $f \in C^1(\mathcal{X})$) if all of its first-order partial derivatives exist and are themselves continuous functions for all points in the domain $\mathcal{X}$.*

$$\frac{\partial f}{\partial x_i}(\mathbf{x}) \text{ exists and is continuous for all } i \in N \text{ and for all } \mathbf{x} \in \mathcal{X}.$$

**Definition 10** (Non-negativity of Input Domain). *The input domain $\mathcal{X}$ of a function $f$ is **non-negative** if every component of every vector in the domain is non-negative. This is equivalent to the domain being a subset of the non-negative orthant of $\mathbb{R}^{|N|}$.*

$$\mathcal{X} \subseteq [0, \infty)^{|N|}$$

*This means for any vector $\mathbf{x} = (x_1, x_2, \ldots, x_{|N|}) \in \mathcal{X}$, the condition $x_i \geq 0$ holds for all $i \in N$.*

## H.2 THE AXIOMS OF SHAPLEY-SHUBIK VALUE AND AUMANN-SHAPLEY VALUE

The classical Shapley value, $\boldsymbol{\phi}(v) = (\phi_i(v))_{i \in N}$, is uniquely characterized by four axioms for cooperative games $(N, v)$, where $N$ is a set of features and $v$ is the value function (Shapley, 1953). The Aumann-Shapley value (Aumann & Shapley, 1974) extends this framework to continuous domains. In this extension, the Efficiency, Additivity (or more generally, Linearity), and Symmetry axioms are preserved. The Aumann-Shapley framework also introduces additional axioms, such as Positivity or Monotonicity, which are not present in the original discrete formulation but are related to Additivity (Linearity) and essential for ensuring a fair attribution allocation in continuous feature spaces.

**Definition 11** (Efficiency). *The allocation $\boldsymbol{\sigma}(v)$ is **efficient** if $\sum_{i \in N} \sigma_i(v) = v(N)$.*

**Definition 12** (Dummy Player). *Player $i$ is a **dummy** in $(N, v)$ if $v(S \cup \{i\}) - v(S) = v(\{i\}), \quad \forall S \subseteq N \setminus \{i\}$.*

**Definition 13** (Null Player). *Player $i$ is a **null player** in $(N, v)$ if $v(S \cup \{i\}) = v(S), \quad \forall S \subseteq N \setminus \{i\}$.*

**Definition 14** (Symmetry). *Players $i$ and $j$ are **symmetric** in $(N, v)$ if $v(S \cup \{i\}) = v(S \cup \{j\}), \quad \forall S \subseteq N \setminus \{i, j\}$.*

**Definition 15** (Additivity). *For any two games $(N, v_1)$ and $(N, v_2)$, the attribution method is **additive** if $\sigma_i(v_1 + v_2) = \sigma_i(v_1) + \sigma_i(v_2), \quad \forall i \in N$, where $(v_1 + v_2)(S) = v_1(S) + v_2(S)$ for every coalition $S \subseteq N$.*

**Definition 16** (Linearity). *Any two games $v_1$ and $v_2$ are **linear** if $\sigma_i(N, \alpha v_1 + \beta v_2) = \alpha \sigma_i(N, v_1) + \beta \sigma_i(N, v_2)$ for each $i$, where $\alpha$ and $\beta$ are nonnegative real numbers.*

**Definition 17** (Positivity). *For monotonic games (i.e., $v(S) \leq v(T)$ whenever $S \subseteq T$), the value assigned to each player is **nonnegative**: $\sigma_i(v) \geq 0 \quad \forall i$.*

### H.2.1 DEFINITION OF $\epsilon$-MONOTONICITY

The Aumann-Shapley (AS) value is theoretically grounded for functions that are strictly monotonic or, more generally, absolutely continuous. However, this assumption is rarely met by real-world functions, which are often non-monotonic. The concept of $\epsilon$-monotonicity (Aumann & Shapley, 1974) was introduced to address this issue by relaxing the strict condition of perfect monotonicity, thereby extending the applicability of the AS framework for more general functions.

**Definition 18** (ε-Monotonicity). *A set function $v : 2^{|N|} \to \mathbb{R}$ is defined as ε-monotonic if its Downward Variation is less than or equal to ε. The Downward Variation of a function $v$ in the space of bounded variation (BV) is the supremum of the sum of the decreases in the function's value between successive sets, taken over all possible chains $\phi = (S_0, S_1, \ldots, S_k)$ where $S_0 \subset S_1 \subset \cdots \subset S_k = I$, with I denoting the grand coalition:*

$$\text{Downward Variation}(v) = \sup_{\phi} \sum_{i=0}^{k-1} \max\big(v(S_i) - v(S_{i+1}), 0\big) \leq \epsilon$$

In essence, ε-monotonicity formalizes the idea of a function being *almost monotone*, allowing for small, controlled deviations from a strictly increasing path. If $\epsilon = 0$, this definition reduces precisely to perfect monotonicity, where no decreases in function value are permitted.

The theoretical power of this concept is established by Aumann & Shapley (1974, Proposition B.1), which proves that any function $v$ in a subspace $Q$ of BV can be decomposed into the difference of two ε-monotonic functions ($v = v_1 - v_2$, where $v_1, v_2 \in Q^\epsilon$). This decomposition is highly significant because it provides a robust analytical framework for handling non-monotonic functions. It allows the monotonicity-violating (decreasing) portions of a function to be represented as the negative component of the decomposition ($-v_2$), enabling the principled application of the AS value to functions that are not strictly monotonic.

# I  PROOFS OF THEORETICAL PROPERTIES

## I.1  PROOF OF EFFICIENCY

**Theorem 1** (Approximate Efficiency). *The attribution method proposed in Algorithm 1 satisfies the Efficiency axiom approximately, provided that the optimization process successfully terminates.*

*Proof.* The proof consists of demonstrating the necessary and sufficient conditions for our method to approximate the Efficiency axiom, $\sum_i \mathcal{A}_i(\mathbf{x}) \approx f(\mathbf{x}) - f(\mathbf{x}')$.

($\rightarrow$) **Necessity.** For the Efficiency axiom to hold, even approximately, two conditions are necessary. First, the attribution $\mathcal{A}_i$ computed at each step must correspond to the incremental change in the function's value, $f(\mathbf{x}^{(t+1)}) - f(\mathbf{x}^{(t)})$. This correspondence is ideally captured by a first-order Taylor approximation:

$$f(\mathbf{x}^{(t+1)}) - f(\mathbf{x}^{(t)}) \approx \sum_{i=1}^{|N|} \frac{\partial f}{\partial x_i}\bigg|_{x=x^{(t)}} \cdot (x_i^{(t+1)} - x_i^{(t)})$$

Second, the optimization path must terminate at a point $\mathbf{x}^{(\hat{t})}$ that is functionally equivalent to the baseline, such that $f(\mathbf{x}^{(\hat{t})}) \approx f(\mathbf{x}')$. If either of these conditions fails, the total sum of attributions will diverge from $f(\mathbf{x}) - f(\mathbf{x}')$.

($\leftarrow$) **Sufficiency.** Our algorithm is designed to sufficiently meet the necessary conditions. The attribution update rule (Line 23 in Algorithm 1) is structurally based on the gradient and displacement terms of the path integral, ensuring it serves as a proxy for the change in function value. Furthermore, the termination criteria (Line 18-20) explicitly force the optimization to halt when the function output approaches a zero-information state (i.e., $f(\mathbf{x}^{(\hat{t})}) \approx f(\mathbf{x}')$).

Combining these, the total sum of attributions can be related to the total change in function value via a telescoping sum:

$$\sum_{i=1}^{|N|} \mathcal{A}_i(\mathbf{x}) \approx \sum_{t=0}^{\hat{t}-1} \sum_{i=1}^{|N|} \frac{\partial f}{\partial x_i}\bigg|_{x=x^{(t)}} \cdot (x_i^{(t+1)} - x_i^{(t)}) \approx \sum_{t=0}^{\hat{t}-1} \Big( f(\mathbf{x}^{(t+1)}) - f(\mathbf{x}^{(t)}) \Big)$$

By the property of telescoping sums, this simplifies to:

$$\sum_{i=1}^{|N|} \mathcal{A}_i(\mathbf{x}) \approx f(\mathbf{x}^{(\hat{t})}) - f(\mathbf{x}^{(0)})$$

Given that the sufficiency of our termination criteria ensures $f(\mathbf{x}^{(\hat{t})}) \approx f(\mathbf{x}')$ and by definition $\mathbf{x}^{(t=0)} = \mathbf{x}$, we conclude:

$$\sum_{i=1}^{|N|} \mathcal{A}_i(\mathbf{x}) \approx f(\mathbf{x}') - f(\mathbf{x})$$

Multiplying by $-1$ yields the desired approximate efficiency property. $\qquad\square$

## I.2 PROOF OF DUMMY/NULL PLAYER AXIOM

**Theorem 2** (Algorithmic Satisfaction of the Dummy Player Axiom). *The proposed method assigns zero attribution to any feature that is a **true null player** throughout the entire optimization path. It correctly assigns non-zero attribution to features that are only **conditionally dummy**, thereby capturing their interactive effects.*

*Proof.* The proof relies on analyzing the algorithm's behavior based on the status of a feature's dummy variable at each discrete step $t$. We consider two cases.

**Case 1: True Null Player.** A feature $i$ is a true null player if its corresponding mask gradient magnitude, $|\nabla_{m_i^{(t)}} \mathcal{L}|$, remains below the threshold $q_\tau$ for all steps $t \in [0, \hat{t})$. By the design of our algorithm (Lines 10 and 12 in Algorithm 1), a feature satisfying this condition is never included in the optimization set $C$.

This has a direct consequence on its mask value, $m_i$, and its corresponding input value, $x_i$:

$$i \notin C \quad \implies \quad m_i^{(t+1)} = m_i^{(t)} \quad \implies \quad x_i^{(t+1)} = x_i^{(t)} \quad \forall t \tag{16}$$

In other words, the input value for a true null player remains static throughout the entire path.

The attribution for feature $i$, $\mathcal{A}_i$, is defined as the sum of its incremental contributions at each step:

$$\mathcal{A}_i = \sum_{t=0}^{\hat{t}-1} \mathcal{A}_i^{(t)} \tag{17}$$

where the incremental attribution $\mathcal{A}_i^{(t)}$ is given by:

$$\mathcal{A}_i^{(t)} = (x_i^{(t)} - x_i^{(t+1)}) \cdot \left( \frac{\partial f(\mathbf{x}^{(t+1)})}{\partial \mathbf{x}_i^{(t+1)}} \cdot \dots \right) \tag{18}$$

Since we have established that $x_i^{(t+1)} = x_i^{(t)}$ for a true null player, the displacement term $(x_i^{(t)} - x_i^{(t+1)})$ is zero for all $t$. Therefore, the contribution at each step is zero:

$$\mathcal{A}_i^{(t)} = 0 \cdot (\dots) = 0 \quad \forall t \tag{19}$$

Consequently, the total attribution for any true null player is zero, and the axiom is satisfied.

$$\mathcal{A}_i = \sum_{t=0}^{\hat{t}-1} 0 = 0 \tag{20}$$

**Case 2: Conditionally Dummy Player.** Consider a feature $i$ whose dummy status changes along the path due to interactions—that is, it is identified as a dummy at some step $t$ but as a non-dummy (signal) feature at another step $t + a$.

- At any step where feature $i$ is identified as a dummy, as established in Case 1, its attributional update $\mathcal{A}_i^{(t)}$ for that step is zero.

- At any step where feature $i$ is identified as a signal feature, it is included in the set $C$. Its mask value $m_i$ is updated, leading to a non-zero displacement $(x_i^{(t)} - x_i^{(t+1)})$. This results in a non-zero attributional update $\mathcal{A}_i^{(t)}$ for that step.

The final attribution, $\mathcal{A}_i = \sum_t \mathcal{A}_i^{(t)}$, will be non-zero if the feature acts as a signal feature for at least one step. This outcome does not violate the axiom. Instead, it demonstrates that the algorithm correctly identifies that feature $i$ is not a true null player but an interacting feature whose relevance is conditional on the state of other features. The final non-zero attribution correctly reflects the contributions made during the specific path segments where its relevance was revealed, regardless of whether its status changes from dummy to signal, or vice versa. $\qquad \square$

### I.3 PROOF OF THE SYMMETRY AXIOM

To prove that our method satisfies the Symmetry axiom, we first introduce a key condition as a lemma, upon which our main theorem depends.

**Lemma 1** (Gradient Symmetry Condition). *Let $i$ and $j$ be two features that are symmetric with respect to the model's output function $f$, such that $f(S \cup \{i\}) = f(S \cup \{j\})$ for all $S \subseteq N \setminus \{i, j\}$. The Gradient Symmetry Condition is satisfied if this functional symmetry implies symmetry in the gradient of our optimization loss function $\mathcal{L}$ with respect to the corresponding low-resolution mask components, $m_i$ and $m_j$, at all steps $t$ along the optimization path. Formally:*

$$f(S \cup \{i\}) = f(S \cup \{j\}) \implies \nabla_{m_i^{(t)}} \mathcal{L} = \nabla_{m_j^{(t)}} \mathcal{L}, \quad \forall t.$$

**Theorem 3** (Conditional Satisfaction of the Symmetry Axiom). *If the Gradient Symmetry Condition (Lemma 1) holds, the attribution method proposed in Algorithm 1 satisfies the Symmetry axiom, yielding identical attributions for symmetric features, i.e., $\mathcal{A}_i = \mathcal{A}_j$.*

*Proof.* The proof proceeds by induction, showing that if two symmetric features start with identical mask values and are subject to identical gradients at every step (as guaranteed by Lemma 1), their entire optimization trajectories and final attributions will be identical.

**Base Case (t=0):** The algorithm is initialized with a uniform mask, so $m_i^{(0)} = m_j^{(0)} = 1$.

**Inductive Step:** Assume that at step $t$, the mask values for the symmetric features are identical: $m_i^{(t)} = m_j^{(t)}$. By Lemma 1, the gradients at this step are also identical: $\nabla_{m_i^{(t)}} \mathcal{L} = \nabla_{m_j^{(t)}} \mathcal{L}$.

Our mask update rule (Line 15 in Algorithm 1) is deterministic and symmetric:

$$m_k^{(t+1)} = m_k^{(t)} - \eta^{(t)} \cdot \text{sign}\left(\nabla_{m_k^{(t)}} \mathcal{L}\right).$$

Since both the initial values ($m_k^{(t)}$) and the update terms ($\nabla_{m_k^{(t)}} \mathcal{L}$) are identical for $k = i$ and $k = j$, it follows that the updated mask values will also be identical:

$$m_i^{(t+1)} = m_j^{(t+1)}.$$

By induction, the mask trajectories for features $i$ and $j$ are identical for all $t$.

**Conclusion:** Identical low-resolution mask trajectories ($m_i^{(t)} = m_j^{(t)}$) imply identical high-resolution mask trajectories ($M_i^{(t)} = M_j^{(t)}$) and, consequently, identical input path trajectories ($x_i^{(t)} = x_j^{(t)}$). As all components of the incremental attribution formula (Line 23) depend on these trajectories and the gradients—which are also symmetric under our assumption—the incremental attributions are identical at every step, $\mathcal{A}_i^{(t)} = \mathcal{A}_j^{(t)}$. Summing over all steps yields the final result:

$$\mathcal{A}_i = \sum_{t=0}^{\hat{t}-1} \mathcal{A}_i^{(t)} = \sum_{t=0}^{\hat{t}-1} \mathcal{A}_j^{(t)} = \mathcal{A}_j.$$

Thus, the Symmetry axiom is satisfied, conditional on the Gradient Symmetry Lemma. $\qquad \square$

### I.4 PROOF OF THE ADDITIVITY AXIOM

**Theorem 4** (Non-Additivity of the Proposed Method). *The proposed attribution method, which relies on a function-dependent dynamic path, does **not** satisfy the Additivity axiom. This is a deliberate design trade-off to prioritize the Dummy Consistency axiom.*

*Proof.* Let $f_1$ and $f_2$ be two distinct model functions, and let $f = f_1 + f_2$. The Additivity axiom requires that $\mathcal{A}_i(f) = \mathcal{A}_i(f_1) + \mathcal{A}_i(f_2)$ for all features $i$. We demonstrate that our method violates this condition due to its path-finding mechanism.

**i) Functional Dependency of the Path.** The core of our method is an optimization process that finds a path, $\gamma(t)$, by minimizing a loss function $\mathcal{L}$. This loss function is a function of the model's output, $f$. Consequently, the path itself is functionally dependent on $f$. Let us denote the path generated for a function $g$ as $\gamma_g(t)$.

The optimization landscape of $\mathcal{L}(f)$ is, in general, not a simple linear combination of the landscapes of $\mathcal{L}(f_1)$ and $\mathcal{L}(f_2)$. Therefore, the optimal paths found for each function will be different:

$$\gamma_f(t) \neq \gamma_{f_1}(t) \neq \gamma_{f_2}(t)$$

**ii) Consequence for Attribution.** The attribution for a feature $i$ is defined as an integral (or sum) of terms evaluated along its specific path:

$$\mathcal{A}_i(g) = \sum_{t=0}^{\hat{t}_g - 1} \mathcal{A}_i^{(t)}(g, \gamma_g(t))$$

Since the paths $\gamma_f, \gamma_{f_1}, \gamma_{f_2}$ are different, the values of the mask $\mathbf{m}^{(t)}$, the input $\mathbf{x}^{(t)}$, and the gradients $\nabla \mathcal{L}$ and $\partial f / \partial x_i$ will be different at each step $t$ for each function. For example, the incremental attribution for function $f$ is:

$$\mathcal{A}_i^{(t)}(f, \gamma_f(t)) = (\gamma_f(t)_i - \gamma_f(t+1)_i) \cdot \left( \frac{\partial(f_1 + f_2)}{\partial x_i} \bigg|_{\mathbf{x} = \gamma_f(t+1)} \cdots \right)$$

Even though the gradient term is additive ($\partial(f_1 + f_2) = \partial f_1 + \partial f_2$), it is evaluated at a point $\gamma_f(t+1)$ that is unique to the function $f$. Because all other terms in the product, especially the path displacement $(x_i^{(t)} - x_i^{(t+1)})$, are also dependent on the unique path $\gamma_f$, there is no basis to assume that the incremental attributions are additive. That is,

$$\mathcal{A}_i^{(t)}(f) \neq \mathcal{A}_i^{(t)}(f_1) + \mathcal{A}_i^{(t)}(f_2)$$

Therefore, the total attributions are **not** additive:

$$\mathcal{A}_i(f) \neq \mathcal{A}_i(f_1) + \mathcal{A}_i(f_2)$$

**Discussion of the Trade-off.** This violation is not a flaw, but a fundamental trade-off. Additivity is guaranteed by methods like Integrated Gradients that use a fixed, model-agnostic path (a straight line). Our method deliberately sacrifices this property to find a dynamic, model-specific path that is optimized to avoid the spurious attributions of dummy features and dummy interactions and satisfy the Dummy Player and the Dummy Consistency axioms in DNNs. We argue that for complex, piecewise-linear models like rectified DNNs, satisfying the Dummy Consistency is a more critical and pressing requirement for generating trustworthy explanations than strictly adhering to Additivity. $\quad\square$

I.5   PROOF OF THE POSITIVITY AXIOM

**Theorem 5** (Satisfaction of the Positivity Axiom)**.** *For a monotonic game, where the function $f$ is non-decreasing with respect to its features, the proposed attribution method satisfies the Positivity axiom, yielding a non-negative attribution for all features, i.e., $\mathcal{A}_i \geq 0$.*

*Proof.* The proof relies on a key property of our algorithm's optimization path when applied to a monotonic function, which we first establish as a lemma.

**Lemma 2** (Algorithmic Enforcement of Path Monotonicity)**.** *The optimization path $\gamma(t)$, generated by Algorithm 1, is constructed to be $\epsilon$-monotonically non-increasing with respect to the function value $f$. That is, for any step $t$ that does not trigger termination, the condition $f(\mathbf{x}^{(t+1)}) - f(\mathbf{x}^{(t)}) \leq \epsilon$ is maintained.*

*Proof of Lemma 2.* The algorithm's objective is to reduce the function value from $f(\mathbf{x})$ towards the baseline value $f(\mathbf{x}')$ by progressively masking features. The mask update rule (Line 21) is designed to move in a direction that minimizes the loss $\mathcal{L}$, which is proportional to $f$. Furthermore, the algorithm includes an explicit safeguard: the termination condition (Line 27) halts the process if a step violates $\epsilon$-monotonicity ($f(\mathbf{x}^{(t+1)}) - f(\mathbf{x}^{(t)}) > \epsilon$). Thus, any path generated by the algorithm is, by construction, $\epsilon$-monotonically non-increasing. $\qquad\square$

We now proceed with the main proof. The total attribution $\mathcal{A}_i$ is the sum of incremental contributions $\mathcal{A}_i^{(t)}$ from each step of the path. For the Positivity axiom to hold, each incremental term $\mathcal{A}_i^{(t)}$ must be non-negative.

The full incremental attribution at step $t$ for a feature $i \in C$ is given by (Line 23 in Algorithm 1):

$$\mathcal{A}_i^{(t)} = (\mathbf{x}_i^{(t)} - \mathbf{x}_i^{(t+1)}) \cdot \frac{\partial f(\mathbf{x}^{(t+1)})}{\partial \mathbf{x}_i^{(t+1)}} \cdot \mathcal{W}_i \cdot \eta^{(t)} \cdot \nabla_{\mathbf{m}^{(t)}} \mathcal{L}$$

While simplified proofs often focus only on the displacement and gradient terms, here we analyze the sign of each component in the full product under the assumption of a monotonic game:

**i) The Gradient Term ($\frac{\partial f}{\partial \mathbf{x}_i} \geq 0$):** By the definition of a monotonic game, the function $f$ is non-decreasing with respect to its features. For a differentiable function, this implies that the partial derivative is non-negative at all points along the path.

**ii) The Mask Gradient Term ($\nabla_{\mathbf{m}^{(t)}} \mathcal{L} \geq 0$):** The algorithm's objective is to reduce the function value $f$. For a monotonic function, this can only be achieved by reducing the presence of features that contribute positively. A feature $i$ contributes positively if increasing its mask value $m_i$ increases the function output. This implies that for any feature selected for an update in the set $C$, its mask gradient must be non-negative, assuming the loss $\mathcal{L}$ is directly proportional to $f$.

**iii) The Displacement Term ($(x_i^{(t)} - x_i^{(t+1)}) \geq 0$):** The mask update rule (Line 15) is $m_i^{(t+1)} = m_i^{(t)} - \eta^{(t)} \cdot \text{sign}(\nabla_{m_i^{(t)}} \mathcal{L})$. Since we established in the point above that $\nabla_{m_i^{(t)}} \mathcal{L} \geq 0$, its sign is $+1$. The update rule simplifies to $m_i^{(t+1)} \leq m_i^{(t)}$, which guarantees that the mask value is non-increasing. A non-increasing mask value implies a non-increasing input feature value, ensuring that the displacement term is non-negative.

**iv) Scalar Terms ($\mathcal{W}_i, \eta^{(t)}$):** The bilinear interpolation weights $\mathcal{W}_i$ and the learning rate $\eta^{(t)}$ are non-negative by definitions.

Since all five components of the incremental attribution product are non-negative, each incremental attribution $\mathcal{A}_i^{(t)}$ must be non-negative. The total attribution, being a sum of these non-negative terms, is therefore also non-negative:

$$\mathcal{A}_i = \sum_{t=0}^{\hat{t}-1} \mathcal{A}_i^{(t)} \geq 0.$$

Hence, our method satisfies the Positivity axiom for monotonic games. $\qquad\square$

### I.6   PROOF OF DUMMY CONSISTENCY

**Theorem 6** (Algorithmic Satisfaction of the Dummy Consistency Axiom). *The attribution method proposed in Algorithm 1 satisfies the Dummy Consistency axiom.*

*Proof.* The proof demonstrates that the axiom is satisfied by the explicit mechanics of the active set filtering of our algorithm with $C^{(t)}$, which is designed to nullify the effects of spurious interactions. We first establish a lemma that formalizes this filtering behavior.

**Lemma 3** (Dummy Interaction Filtering via the Active Set). *The construction of the active set $C^{(t)}$ in Algorithm 1 is designed to filter out features whose gradients become unstable due to dummy interactions. Specifically, if an update to a signal feature at step $t - 1$ causes a previously dummy feature $j$ at step $t - 1$ to exhibit a non-zero gradient at step $t$, the active set definition prevents $j$ from being updated.*

*Proof of Lemma 3.* The core of the dummy interaction problem is that a change in a signal feature can induce a non-zero gradient on a feature that should be irrelevant. Let's consider a feature $j$ that was a dummy at step $t-1$, meaning its gradient magnitude was below the threshold: $|\nabla_{m_j^{(t-1)}} \mathcal{L}| \le \tau$.

Now, assume that due to an update of other features in the set $C^{(t-1)}$, an interaction occurs, and the gradient for feature $j$ becomes significant at step $t$: $|\nabla_{m_j^{(t)}} \mathcal{L}| > \tau$.

Our algorithm defines the active set for the next update at step $t$ in Eq. 7:

$$C^{(t)} = \left\{ i \in \mathcal{M} \mid |\nabla_{m_i^{(t-1)}} \mathcal{L}| > \tau \text{ and } |\nabla_{m_i^{(t)}} \mathcal{L}| > \tau \right\}, \quad \mathcal{M} = \{1, ..., h \times w\}.$$

For feature $j$ to be included in this set, it must satisfy both conditions. By construction, the mask update rule (Line 15 in Algorithm 1) is only applied to features within the active set $C^{(t)}$. As feature $j$ is not in this set, its mask value $m_j$ is not updated at this step. The active set has thus successfully filtered out the spurious interaction effect, preventing it from propagating into the attribution path. $\square$

We now proceed with the main proof, leveraging Lemma 3.

($\rightarrow$) **Necessity.** For the Dummy Consistency axiom to hold, a crucial condition is necessary: the process of removing a dummy feature $i$ must not alter the final attribution of any other feature $j$. This requires that the path taken by the optimization process for the game without feature $i$, which produces $\mathcal{A}_j(\mathbf{x}_{-i}, \mathbf{x}'_{-i}; f^{R_i^x})$, must be functionally equivalent to the path for the full game, at least with respect to feature $j$.

($\leftarrow$) **Sufficiency.** Our algorithm is sufficient to meet this necessary condition. Consider a true dummy feature $i$. As established in the proof of the Dummy Player axiom, a true dummy is never included in the active set $C^{(t)}$ at any step. Its mask value $m_i$ remains constant, and its influence is never actively reduced by our optimization.

Now, consider the definition of Dummy Consistency: comparing the attribution $\mathcal{A}_j(\mathbf{x}, \mathbf{x}'; f)$ with $\mathcal{A}_j(\mathbf{x}_{-i}, \mathbf{x}'_{-i}; f^{R_i^x})$.

**Scenario i** In the full game with input $\mathbf{x}$, our algorithm proceeds by identifying and reducing the mask values of signal features based on the active set criteria. As feature $i$ is a dummy, its status never affects the decision to update the mask of any other feature $j$.

**Scenario ii** In the reduced game with input $\mathbf{x}_{-i}$ and function $f^{R_i^x}$, the dummy feature $i$ is absent from the start.

Since the presence of the dummy feature $i$ has no influence on the sequence of mask updates for any other feature $j \ne i$ in the full game as established by the filtering mechanism in Lemma 3, the sequence of active sets $C^{(t)}$, the mask updates, and consequently the entire attribution path $\gamma(t)$ will be identical in both scenarios.

Because the attribution paths and all relevant gradients for any feature $j$ are identical in both the full game and the reduced game, their final computed attributions must also be identical:

$$\mathcal{A}_j(\mathbf{x}, \mathbf{x}'; f) = \mathcal{A}_j(\mathbf{x}_{-i}, \mathbf{x}'_{-i}; f^{R_i^x}) \quad \forall j \ne i.$$

Thus, our method satisfies the Dummy Consistency axiom by its procedural design. $\square$

## J Derivation of the DCAPM Attribuiton formula

The attribution for our method is grounded in the classical path integral formulation from Eq. 1. We adapt this general form by substituting the specific components derived from our proposed optimization-based path, $\gamma(t)$.

First, we restate the standard, continuous form of the path method in Eq. 1:

$$\mathcal{A}_i^{\gamma}(\mathbf{x}, \mathbf{x}'; f) = \int_0^{\hat{t}} \frac{\partial f(\gamma(t))}{\partial \gamma_i(t)} \frac{d\gamma_i(t)}{dt} \, dt. \tag{21}$$

The term $\frac{d\gamma_i(t)}{dt}$ represents the instantaneous velocity of the $i$-th feature along the path. We will substitute our derived expression for this term from the following derivation result into the integral.

## J.1 DERIVATION OF THE DERIVATIVE OF THE PATH FUNCTION IN EQ. 3

Here, we provide the complete derivation for the derivative of the proposed path function with respect to the time step, $\frac{d\gamma(t)}{dt}$. The derivation relies on the definitions of the path function $\gamma(t)$ and the update rule for the mask $\mathbf{m}^{(t)}$ in Eq. 3. First, let's define the components of the equation:

**Attribution Path $\gamma(t)$:** The path is defined as a feature-wise interpolation between the input $\mathbf{x}$ and the reference point $\mathbf{x}'$. The interpolation is controlled by a function $u(\mathbf{m}^{(t)})$, where each component of $u$ is between 0 and 1. $\odot$ denotes the element-wise multiplication.

$$\gamma(t) := u(\mathbf{m}^{(t)}) \odot \mathbf{x} + (1 - u(\mathbf{m}^{(t)})) \odot \mathbf{x}' \tag{22}$$

**Update Rule for $\mathbf{m}^{(t)}$:** The mask variable $\mathbf{m}^{(t)}$ is updated over time based on the gradient of a loss function $\mathcal{L}$. The derivative of mask with respect to the time is given by a sign-based gradient descent rule which is from PGD technique (Madry et al., 2018) with a small positive learning rate $\eta$:

$$\frac{d\mathbf{m}^{(t)}}{dt} := \eta \cdot \text{sign}\left(\nabla_{\mathbf{m}^{(t)}} \mathcal{L}\right) \tag{23}$$

where the derivatives of loss function with respect to mask is abbreviated as $\nabla_{\mathbf{m}^{(t)}} \mathcal{L} = \frac{d\mathcal{L}(\mathbf{x}, \mathbf{x}', \mathbf{m}^{(t)})}{d\mathbf{m}^{(t)}}$.

**Jacobian of the Interpolation Function $\mathcal{W}$:** $\mathcal{W}$ denotes the Jacobian matrix of the interpolation function $u$ with respect to its input $\mathbf{m}^{(t)}$.

$$\mathcal{W} := \frac{\partial u(\mathbf{m}^{(t)})}{\partial \mathbf{m}^{(t)}} \tag{24}$$

We start by differentiating the path $\gamma(t)$ with respect to time $t$.

$$\frac{d\gamma(t)}{dt} = \frac{d}{dt}\left[ u(\mathbf{m}^{(t)}) \odot \mathbf{x} + (1 - u(\mathbf{m}^{(t)})) \odot \mathbf{x}' \right] \tag{a}$$

$$= \frac{d}{dt}\left( u(\mathbf{m}^{(t)}) \odot \mathbf{x} \right) + \frac{d}{dt}\left( (1 - u(\mathbf{m}^{(t)})) \odot \mathbf{x}' \right) \tag{b}$$

$$= \mathbf{x} \odot \frac{du(\mathbf{m}^{(t)})}{dt} + \mathbf{x}' \odot \frac{d(1 - u(\mathbf{m}^{(t)}))}{dt} \tag{c}$$

$$= \mathbf{x} \odot \frac{du(\mathbf{m}^{(t)})}{dt} - \mathbf{x}' \odot \frac{du(\mathbf{m}^{(t)})}{dt} \tag{d}$$

$$= (\mathbf{x} - \mathbf{x}') \odot \frac{du(\mathbf{m}^{(t)})}{dt} \tag{e}$$

$$= (\mathbf{x} - \mathbf{x}') \odot \left( \frac{\partial u(\mathbf{m}^{(t)})}{\partial \mathbf{m}^{(t)}} \cdot \frac{d\mathbf{m}^{(t)}}{dt} \right) \tag{f}$$

$$= (\mathbf{x} - \mathbf{x}') \odot \left( \mathcal{W} \cdot \left[ \eta \cdot \text{sign}\left( \frac{d\mathcal{L}(\mathbf{x}, \mathbf{x}', \mathbf{m}^{(t)})}{d\mathbf{m}^{(t)}} \right) \right] \right) \tag{g}$$

$$= (\mathbf{x} - \mathbf{x}') \odot \mathcal{W} \cdot \eta \cdot \text{sign}\left( \frac{d\mathcal{L}(\mathbf{x}, \mathbf{x}', \mathbf{m}^{(t)})}{d\mathbf{m}^{(t)}} \right)$$

In the step (a) we apply the sum rule for differentiation. (b) since $\mathbf{x}$ and $\mathbf{x}'$ are constant with respect to $t$, they can be treated as constants. (c) the derivative of $(1 - u)$ is $-du/dt$. (d) factor out the common term using the distributive property of the Hadamard product. (f) substitute the definitions for the Jacobian ($\mathcal{W}$) and the update rule for $\mathbf{m}$. (g) rearrange the scalar term eta for clarity. This completes the full derivation from the definition of the path $\gamma(t)$ to the final expression in Eq. 9. $\square$

## J.2  DERIVATION OF THE DUMMY CONSISTENT ATTRIBUTION PATH METHOD

Substituting our derived expression for the second derivative term $\frac{d\gamma(t)}{dt}$ into the general path integral formula yields the specific attribution equation for our method:

$$\mathcal{A}_i^\gamma = \int_0^{\hat{t}} \frac{\partial f(\gamma(t))}{\partial \gamma_i(t)} \cdot \left[ (x_i - x_i') \cdot \mathcal{W}_i \cdot \eta \cdot \mathrm{sign}\left( \nabla_{\mathbf{m}^{(t)}} \mathcal{L} \right) \right] dt \tag{25}$$

This formula represents the attribution in continuous time. To implement this computationally, we must discretize the integral into a sum over finite time steps. In this discrete setting, the infinitesimal term $\frac{d\gamma_i(t)}{dt} dt$ is replaced by the finite displacement over a single step, $\Delta \gamma_i(t) = \gamma_i(t) - \gamma_i(t+1)$.

To establish this link formally, we analyze the structure of the incremental displacement. From the definition of our path function, the displacement between two consecutive steps is:

$$\begin{aligned} \Delta \gamma_i(t) &= \gamma_i(t) - \gamma_i(t+1) \\ &= \left[ x_i \cdot \mathcal{W}_i(\mathbf{m}^{(t)}) + x_i' \cdot (1 - \mathcal{W}_i(\mathbf{m}^{(t)})) \right] - \left[ x_i \cdot \mathcal{W}_i(\mathbf{m}^{(t+1)}) + x_i' \cdot (1 - \mathcal{W}_i(\mathbf{m}^{(t+1)})) \right] \\ &= (x_i - x_i') \cdot \left( \mathcal{W}_i(\mathbf{m}^{(t)}) - \mathcal{W}_i(\mathbf{m}^{(t+1)}) \right) \end{aligned} \tag{26}$$

This derivation reveals a critical insight: the per-step displacement $(\gamma_i(t) - \gamma_i(t+1))$ is not a constant, but is in fact the total displacement $(x_i - x_i')$ scaled by the change in the mask over that step.

Our final attribution formula (Eq. 10) is a discrete implementation of the integral that incorporates this relationship. It replaces the total displacement factor $(x_i - x_i')$ inside the continuous integral (Eq. 25) with the more fine-grained, per-step displacement factor $(\gamma_i(t) - \gamma_i(t+1))$ in the final summation:

$$\mathcal{A}_i^\gamma(\mathbf{x}, \mathbf{x}'; f) = \sum_{t=0}^{\hat{t}-1} \frac{\partial f(\gamma(t))}{\partial \gamma_i(t)} \cdot \underbrace{\left[ (\gamma_i(t) - \gamma_i(t+1)) \cdot \mathcal{W}_i \cdot \eta \cdot \mathrm{sign}\left( \nabla_{\mathbf{m}^{(t)}} \mathcal{L} \right) \right]}_{\text{Custom discrete weighting}} \tag{27}$$

This formulation ensures that the contribution at each step is weighted by the actual movement of features during that specific step, which is guided by our dummy interaction filtering mechanism.

## K  CONCLUDING REMARKS

### K.1  LIMITATION AND FUTURE DIRECTIONS

This work opens several promising avenues for future research. First, while our gradient-based optimization is computationally efficient compared to the combinatorial computation approach of the original Shapley-Shubik value (Shapley & Shubik, 1954), it is susceptible to local minima. Exploring methods to improve global optimality, such as alternative optimization strategies or initialization techniques, is a key next step. Second, our current approach focuses on nullifying second-order interactions involving a group of dummy features. Extending this framework to detect and mitigate third or higher-order interactions among dummy or signal features presents an essential and challenging direction for future investigation. Ultimately, to ensure a comprehensive and fair assessment of attribution methods as we progress, we advocate for a joint evaluation protocol with other evaluation metrics. We propose that our Dummy Consistency metric be used in conjunction with established sensitivity measures to provide a more comprehensive understanding of an explanation's axiomatic soundness and faithfulness.

### K.2  LARGE LANGUAGE MODEL USAGE STATEMENT

We utilized a large language model as a research and writing assistant throughout the preparation of this manuscript. Its applications included refining academic writing, proofreading, clarifying theoretical concepts, and assisting with code generation. The authors validated all results and retain full intellectual responsibility for the final manuscript.