# OpenReview forum: "Dummy Consistent Path Method for Robust Attribution with Irrelevant Features"
_ICLR.cc/2026/Conference — Submitted to ICLR 2026_

### Official Review · Reviewer_RB8P · 2025-10-20

**Soundness:** 3
**Presentation:** 1
**Contribution:** 3
**Rating:** 4
**Confidence:** 4

**Summary:**

This paper proposes Dummy Consistent Attribution Path Method (DCAPM), a path-based attribution method that rigorously satisfies the Dummy Player and Dummy Consistency axioms. Specifically, DCAPM dynamically constructs an integration path, nullifying spurious attributions from irrelevant feature interactions with dummies to ensure zero attribution at every step, grounded in a principled approximation of the Shapley interaction value.

**Strengths:**

The topic this paper focused on is very important in explainable AI.

**Weaknesses:**

1. The explanation for the " integration path" in introduction section is unclear. Can authors provide any vivid example for explanation? The physical meaning for the mapping $\gamma$ in Definition 1 of path function is missing, making it difficult to understand.

2. Eq. (2) only considers the interaction between two features i and j. How about interactions among features more than 2? In implementation, how to select features i and j? How about its computation complexity? Can authors provide a pseudocode for the proposed method？

3. The physical meaning and the motivation of the loss function in Eq. (3) is missing. Authors should clearly explain this function, as it is the main contribution of this paper.

4. From figure 3, I cannot agree with author's opinion that the proposed method has minimal background noise. However, DGA has less background noise than the proposed method. Thus, this result seem not convincing enough to verify the efficiency of the proposed method.

5. Could the proposed method be applied to transformer-based models and ConvNext model that have achineved sota performance nowadays?

6. Can the proposed method be applied to the dataset with spurious correlations, such as waterbird datasets?

**Questions:**

Please refer to the weaknesses.
If all of my problems are addressed, I will raise my rating.
I like this paper, however, the presentation is poor, especially lack of physical meaning for each equation. Besides, the designed experiments is too simplistic and results seems not convincing enough.

---

> ### Author Response · Authors · 2025-11-26
> **Author Response to Reviewer RB8P (1/2)**
>
> Thank you for the detailed review and constructive feedback on our work. Please find our point-by-point response below, indexed as [C1] through [C6].
>
> **[C1]  “physical meaning of integration path”**
>
> We agree that providing a vivid example and clarifying the physical meaning will significantly enhance understanding. We have revised the manuscript as follows:
>
> - **Clarification of "Integration Path" with physical meaning and vivid example (Figure 2):** We have refined the terminology from *"integration path"* to *"path, a sequence of intermediate inputs evolving from the baseline to the original input (see 2nd row in Figure 2),"* in the Introduction (Line 88) to reduce jargon and clarify the physical meaning and explicitly referenced Figure 2 (2nd row) as a vivid visualization of this concept.
>
> - **Refined Definition of Path Function (Section 2.1):** To address the comment on the *"mapping"* definition, we have updated Definition 1 to clarify the domains and, crucially, added a description of its physical meaning.
>
> > Revised Definition 1 (Path Function): A path function is a mapping $\gamma: [0, \hat{t}] \times \mathcal{X}(N) \times \mathcal{X}(N) \to \mathcal{X}(N)$ for a finite stopping step $\hat{t} < \infty$. For a time step $t \in [0, \hat{t}]$, a baseline $\mathbf{x}' \in \mathcal{X}(N)$, and an input $\mathbf{x} \in \mathcal{X}(N)$, the mapping $\gamma(t; \mathbf{x}, \mathbf{x}')$ generates an intermediate input state. It satisfies continuous and feature-wise non-decreasing boundary conditions: $\gamma(0; \mathbf{x}, \mathbf{x}') = \mathbf{x}'$ (baseline) and $\gamma(\hat{t}; \mathbf{x}, \mathbf{x}') = \mathbf{x}$ (original input).
> >
> **[C2] “role of each term in the loss function”**
>
> We have revised Section 3.1 to explicitly describe the role of each term as follows:
>
> - **First term ($f(\gamma(t))$) as the driving force for feature removal:** By minimizing $f(\gamma(t))$ via gradient descent, the optimization actively identifies and erases features that contribute positively to the target class. As a result, as relevant features are identified and their mask values pushed from 1 to 0 (replaced by the baseline $x'$), the function output $f(\gamma(t))$ decreases monotonically. This creates a path that fades out signal features first.
> - **L1 Regularization Term ($\lambda |1 - u(m^{(t)})|_1$) as the selector for dummy preservation:** This term acts as a resistance force that penalizes the modification of the input (i.e., changing mask from 1 to 0). This term encourages sparsity in the resulting mask, effectively keeping the mask at 1 for features that do not significantly affect the model output (i.e., dummy features).
>
> Consequently, the mask $u(m)$ converges to a state where relevant features are 0 and dummy features remain 1. This isolates the "dummy features" to form the endpoint of our path ($x_{dum}$). We have updated the manuscript (in line 269) to include this detailed physical interpretation, ensuring the motivation behind Eq. 3 is clearly conveyed.
>
> **[C3] “denoising quality of DGA vs. DCAPM in Figure 3”**
>
> We respectfully disagree with the interpretation that DGA has less background noise. We would like to clarify this through a detailed analysis of the Row 1 in Figure 3: DGA’s heatmap assigns non-zero attribution to the background surrounding the tennis ball. It indicates a miss-allocation of attribution into dummy regions. In contrast, DCAPM effectively reduces attributions in irrelevant regions to zero (represented as white). This is objectively supported by Table 1, where DCAPM achieves the best Dummy Consistency score (0.0002 vs. DGA's 0.225 on VGG16), quantitatively demonstrating it is the most effective at eliminating irrelevant attribution.
>
> **[C4] “interaction between features $i$ and $j$ in Eq. 2, more than two cases, computation complexity, pseudocode”**
>
> We utilize the first-order Taylor approximation based on the difference in gradients to compute the Shapley interaction, following Eq. 10 in Chang et al. (2025). Here, features $i$ and $j$ correspond to the set of existing features and absent features at step $t-1$, respectively ($i \in S^{(t-1)}$ and $j \notin S^{(t-1)}$). This formulation interprets the interaction between $i$ and $j$ as the variation in the gradient of feature $j$ resulting from the removal of feature $i$. The higher-order interactions (more than two features) require calculating higher-order derivatives (e.g., Hessian), which significantly increases computational cost. Therefore, we focus on pairwise interactions to maintain computational efficiency while sufficiently capturing the spurious relationships between features. The computational complexity of DCAPM is linear with respect to the number of optimization iterations $T$, i.e., $O(T)$. As detailed in Appendix G.2, empirically, a single image takes approximately 22.02 seconds (for 600 iterations). The detailed pseudocode is provided in Algorithm 1 (lines 9–17) in Appendix B.

---

> ### Author Response · Authors · 2025-11-26
> **Author Response to Reviewer RB8P (2/2)**
>
> **[C5] “SOTA transformer-based models (ConvNext)”**
>
> We are currently conducting the additional experiments on the ConvNeXt model suggested by the reviewer. As these experiments require some computational time, we will update the manuscript with the new results as soon as they are ready.
>
> **[C6] “Can the proposed method be applied to the dataset with spurious correlations?”**
>
> We agree with the reviewer regarding the importance of revealing spurious correlations. To address this, we added an analysis in Appendix C.4, showing that such correlations are identifiable even in the ImageNet dataset, thereby highlighting the practical utility of DCAPM.

---

> > ### Comment · Reviewer_RB8P · 2025-11-27
> > **Response to rebuttal**
> >
> > Thanks for author's response. First, I think authors should consider interactions among more than two variables. Second, where is Appendix C.4? How the proposed method works on the waterbird dataset, which is a very classical dataset with spurious correlations?

---

> ### Author Response · Authors · 2025-12-02
> **Author Response to Reviewer RB8P**
>
> We appreciate the reviewer's feedback.
>
> **1. Interactions among more than two variables**
>
> We acknowledge the theoretical importance of capturing high-order interactions. However, we restricted our scope to pairwise interactions for two primary reasons:
>
> - **Mathematical Consistency:** Our approach utilizes the first-order Taylor approximation framework from Chang et al. (2025). In Piecewise Linear Neural Networks (PLNNs), second- and higher-order derivatives are zero almost everywhere. Thus, within this approximation framework, higher-order interaction terms are negligible.
>
> - **Computational Efficiency:** Calculating high-order interactions increases complexity exponentially. We found that pairwise interactions provide the optimal trade-off between attribution fidelity and computational feasibility.
>
> We have explicitly discussed this in the **Limitations section (Appendix K.1)** and acknowledge the extension of the \textit{Dummy Consistency} axiom to high-order interactions as a valuable direction for future work.
>
> **2. Waterbirds Analysis in Appendix C.5**
>
> The requested section is located in Appendix C.5 of the **revised manuscript**, with all new text highlighted in blue. As suggested, we performed a spurious correlation analysis on the **Waterbirds** dataset. The results demonstrate that DCAPM precisely highlights the spurious correlations learned by the model while successfully filtering out noisy attributions. This confirms DCAPM's superior utility for model debugging.
>
> **3. ConvNeXt experiments [C5]**
>
> As suggested, we evaluated DCAPM against baselines (DGA, IG) on the ConvNeXt-T model. DCAPM demonstrated superior attribution fidelity with a mean DiffID score of $0.4801$, outperforming both DGA ($0.4737$) and IG ($0.2306$). We have added these results to **Appendix C.4**.

---

### Official Review · Reviewer_FCW1 · 2025-10-29

**Soundness:** 2
**Presentation:** 3
**Contribution:** 1
**Rating:** 2
**Confidence:** 3

**Summary:**

This paper aims to improve gradient-based feature attribution methods, such as integrated gradients, that use a path integral to quantify feature attribution values. The authors target so-called "dummy features", which are described as features that do not contribute to the prediction of the explained instance. It is observed that due to evaluating the gradients at different regions of the feature space, such dummy features are attributed non-zero attribution values, since their gradients to not necessarily vanish along the whole path. The authors then propose the "Dummy Consistent Path Method" as a multi-step process: First, a path is constructed along masked inputs, where masks are iteratively constructed by optimizing for sparse masks that significantly drop the payout $f$. Second, the "relevant" features are identified by using a threshold for the partial derivatives obtained at each iteration. Third, the derivative of the path is then used to construct the final attribution method.

**Strengths:**

- This paper identifies the important problem that averaging gradients from different regions (e.g. along the path from $x$ to the baseline $x'$ in IG) may assign contributions to features that were not used by the model at the original data point $x$. As a solution, the authors essentially propose to mask only features, where the masking reveals a significant drop in the prediction, which is novel and interesting.
- Dealing with such dummy features by constraining the computation only to features that were not identified as dummy in the prediction of $x$ (and later on) is interesting
- The paper provides a comprehensive methodology, description and experimental details. It is well structured and clearly presented.

**Weaknesses:**

- One major problem is the motivation: The authors present dummy consistency (Definition 4) as receiving zero attribution when all gradients along the path are zero. Clearly IG satisfies this axiom. In contrast, Figure 1 claims that this axiom is violated. However, as the authors describe themselves (see "spurious interactions"), the non-zero attribution is due to the traversal across other regions of the feature space, where the model indeed uses this feature to predict the output, resulting in a non-zero gradient. Moreover, I would have expected to see DCAPM method here in this figure, as I do not see, how the novel method resolves these issues. In fact, the whole point of evaluating not only the gradients at $x$ (also for simple methods like SmoothGrad) is that we investigate regions "close" to $x$, since gradients are a very local measure, especially in high-dimensional DNNs. Therefore, any path-based method crossing regions, will be confronted with the problem that the gradients (i.e. attribution values) are not necessarily the same at another instance, and features that were not relevant before, could now be relevant.
- As a result of the first point: I think this paper's motivation is rather constructing a path that masks more effectively selected features that are "relevant" to the prediction, since their masking substantially lowers the prediction / payout $f$. As a result, the method will only investigate changes in features that were relevant at the explained instance $x$. But this could have been done without even using an iterative optimization procedure.
- Another major problem is the methodology: As the authors note, that problem of "dummy (in)consistency" arises from the fact that the gradients are evaluated along the path and do not necessarily reflect the gradients at the original input $x$. The authors restrict the update of the (continuous) masks to "important" features identified by large enough gradients, i.e. these features significantly change the prediction, if masked. However, if we assume that the feature space is partitioned by the DNN into regions, where the prediction is linear, then one of the following two scenarios might happen:
  - A: The new masked input is in the same region, i.e. the gradients remain unchanged or
  - B: The new masked input is in a different region, i.e. the gradients can arbitrarily change, meaning that previously feature identified as unimportant (zero gradient) could now become relevant (non-zero gradient). This is due to the fact that relevant features were only taken into account when constructing the mask and not when evaluating the gradients of the model.

  Right now, I do not see, why this attribution method would suffer less from assigning non-zero contributions to features that were not relevant at point $x$. Moreover, I would expect the objective the authors try to optimize would be already resolved by using the most basic method Input $\times$ gradients.

**Questions:**

- Could you explain, what you mean by dummy consistency? In the draft, it seems that this definition is used interchangeably for features that obtain zero attribution, if they are "not relevant" (close to zero gradients) at the explained instance $x$, or whose gradients are zero along the whole path (Definition 4).
- Could you explain, why your method will assign close-to-zero-attributions for features that have close-to-zero-gradients at the explained instance $x$. As far as I understand, you only restrict the mask to "relevant" features, but the model could still predict very differently for the novel masked input.
- Follow-up to the previous question: Is it overall desirable to enforce zero-attributions for features that have zero-attributions at $x$? What if the model is very non-robust and actually uses features for instances that look very much like $x$?
- Could you add your method to the example in Figure 1? Moreover, could you add a simple baseline, like Grad$\times$Input?

---

> ### Author Response · Authors · 2025-11-26
> **Author Response to Reviewer FCW1 (1/2)**
>
> Thank you for the detailed review and constructive feedback on our work. We especially appreciate your insightful question about the necessity of our iterative optimization procedure over simpler static baselines, which is particularly helpful for clarifying our contribution. Please find our point-by-point response below, indexed as [C1] through [C7].
>
> **[C1] “One major problem is the motivation”**
>
> We understand the reviewer's perspective that if a model utilizes a feature along a path, the attribution should reflect that (Faithfulness). However, our work addresses the specific pathology of deep ReLU networks, where this assumption leads to noisy and unreliable explanations.
>
> - **Theoretical Satisfaction vs. Practical Violation (Shattered Gradients):** We agree that if gradients are exactly zero along the path, IG yields zero attribution. However, as noted in the literature (Balduzzi et al., 2017; Montufar et al., 2014) and our paper, deep rectified networks suffer from "Shattered Gradients." Even for irrelevant features for the model's decision, the gradients are rarely zero due to the piecewise-linear nature of the decision boundary. The path methods unknowingly integrate these noisy gradients, assigning importance to features that are functionally irrelevant to the model's classification. This violation is visualized in Figure 1(d).
>
> We also understand the concern that zeroing out gradients might ignore a relevant feature that the model genuinely utilizes.  However, we would like to clarify that DCAPM does not blindly ignore gradients; rather, it acts as a dynamic refinement mechanism to distinguish between transient noise (shattered gradients) and true relevance. Our approach gives every feature the "opportunity" to contribute to the attribution if—and only if—it demonstrates significant sensitivity along the path. Therefore, DCAPM does not ignore valid gradients; it refines the attribution by filtering out weak signals arising from the shattered gradients, while aggregating gradients that correspond to significant (relevant) interactions.
>
> - **Motivation Figure Update:** As requested, we have added DCAPM to Figure 7 in Appendix C.2. It demonstrates that our method successfully avoids these spurious interaction regions, assigning zero attribution to the dummy feature ($x_3$) and recovering consistent attributions for the signal features ($x_1, x_2$), unlike IG.
>
> **[C2] Necessity of “using an iterative optimization procedure”**
>
> We respectfully clarify that heuristic masking or feature localization methods (e.g., RISE, Occlusion) differ fundamentally from our approach. While these methods estimate relevance based on local perturbations around the static input $x$, our path-based method aggregates infinitesimal contributions along a trajectory from a baseline. Crucially, this allows our approach to give every feature the 'opportunity' to contribute to the attribution whenever it becomes relevant along the path. This aggregation is critical because the model relies on different features at various points along the path. Simple non-iterative methods examine features relevant only at $x$, ignoring how feature importance evolves and interacts during the transition. Consequently, static methods cannot guarantee fundamental fairness properties like the Dummy Player and Dummy Consistency axioms. Our iterative optimization is specifically designed to navigate these changing interactions, ensuring a valid path that strictly nullifies irrelevant signals.

---

> ### Author Response · Authors · 2025-11-26
> **Author Response to Reviewer FCW1 (2/2)**
>
> **[C3] “methodology (A and B scenarios) and Input $\times$ Gradient comparison”**
>
> We address the comparison with Input $\times$ Gradient and clarify how our method robustly leverages both single-point gradients (Scenario A) and region transitions (Scenario B).
>
> - **Instability of "Input $\times$ Gradient" (Robustness Issue):** The reviewer suggested "Input $\times$ Gradient" as a potential solution. However, we argue that this baseline is prone to severe robustness issues, similar to the known limitations of raw Saliency Maps. In deep networks, decision boundaries are highly fragmented, making the model output sensitive to small perturbations (adversarial robustness). Therefore, relying solely on the local gradient at $x$ yields unstable attributions. Path-based attribution aggregation is necessary to smooth out these local irregularities.
>
> - **Handling Region Crossings via Progressive Aggregation (Analogy to Watershed Algorithm):** Regarding the concern about crossing linear regions (Scenario B), our iterative optimization handles this naturally through a step-wise mechanism: **i) Step-wise Zeroing:**  At each step $t$, features with near-zero gradients do not move. Mathematically, the term $(\gamma_t - \gamma_{t-1})$ becomes zero, ensuring that no attribution accumulates for these "locally dummy" features at that specific step. **ii) Dynamic Aggregation:**  As the path moves to step $t+1$, it may enter a new linear region where the gradients are independent of the previous step. A feature that was previously inactive may now exhibit a meaningful gradient.
>
> - **Meaning:** This process effectively aggregates attributions starting from features with sufficiently steep gradients. As the path evolves, we progressively capture smaller yet meaningful contributions, unlike a static linear path that blindly aggregates arbitrary gradients.
>
> **[C4] “meaning of dummy consistency”**
>
> We distinguish between the axiomatic definition and our practical identification: Definition 4 states that removing a dummy player (feature with zero marginal contribution) should not change the attribution of others. We do not define dummy features solely based on the gradient at input $\mathbf{x}$. Instead, a feature is identified as a dummy dynamically at each step $t$ if its contribution to the loss function $\mathcal{L}$ (Eq. 3) falls below a threshold $\tau_g$. This ensures we capture features that are irrelevant throughout the path integration process, not just at the single instance $\mathbf{x}$.
>
> **[C5] “close-to-zero attributions”**
>
> The reviewer correctly pointed out that restricting masks based solely on $\mathbf{x}$ would be risky. However, our method does not rely on a static mask derived from $\mathbf{x}$. As described in Sec 3.1, we optimize the mask $m^{(t)}$ iteratively. The first term in Eq. 3 ($f(x \odot u(m^{(t)}) + \dots)$) explicitly ensures that the model's prediction is maintained as the path evolves.
>
> **[C6] “when restricting the mask to relevant features, then the model predict differently to the masked input”**
>
> We use the fixed predicted class of the original input $\mathbf{x}$ as the target throughout the entire path optimization. As shown in the logit graph in Figure 2 (Row 1), our method ensures that the function output $f(\gamma(t))$ transitions smoothly from the input prediction to the baseline and reduces monotonically as information is removed.
>
> **[C7] “add DCAPM and Input $\times$ Grad to Figure 1”**
>
> We have updated Figure 1 and Appendix C.2 to include these results.

---

### Official Review · Reviewer_N75c · 2025-10-31

**Soundness:** 3
**Presentation:** 2
**Contribution:** 3
**Rating:** 6
**Confidence:** 3

**Summary:**

The authors present the Dummy Consistent Attribution Path Method (DCAPM), which optimizes attribution paths with a mask-based approach while explicitly "blocking" the interaction between dummy and signal features at each step. The method is designed to satisfy the Dummy Player and Dummy Consistency axioms, thus reducing errors caused by spurious interactions, commonly referred to as "fragmented gradients." The manuscript provides method derivations, evaluation metrics, and comparison experiments.

**Strengths:**

1.	Clear Focus on an Important Issue: The paper addresses Dummy Consistency, an aspect of attribution stability that has been relatively under-discussed. The authors elevate this principle to the level of method design, offering both theoretical and practical significance (see Sec. 2.2 and the overall motivation).
2.	Intuitive and Interpretable Method: The use of low-resolution masks and optimization to produce dynamic baselines is clearly articulated. It provides a strategy for "finding clean baselines per sample and avoiding noise gradient accumulation in attribution" (see Sec. 3.1–3.3).
3.	Introduction of a New Metric: The authors propose the Dummy-Consistency metric (Eq. 11) to quantify the satisfaction of the axioms, which concretizes the concept of "robustness."

**Weaknesses:**

1.	Questionable Table 1 Results (DiffID):
For example （not only this one）, Table 1 shows a result of DiffID = 0.978 ± 6.41 for InceptionV3, where the standard deviation is much larger than the mean. DiffID measures the difference in attribution values between the most significant inserted pixels and the least significant deleted pixels. While a higher DiffID indicates the method's ability to identify important features, the large error margin significantly affects the interpretability and trustworthiness of the analysis. This issue is also present in other experiments in the appendix. Please address this inconsistency and clarify the statistical reporting.

2.	Lack of Empirical Statistical Analysis for Dummy Consistency:
The paper emphasizes that a primary contribution is enforcing the Dummy Consistency axiom during path optimization (as discussed in Sec. 3), where spurious interactions between features are supposed to be removed. Eq. (6) is used to quantify this, and the goal is to make it zero. However, it does not show a empirical statistical analysis to support these claims. The appendix includes a proof, but would benefit from including relevant statistical evidence for the empirical results, regarding the effectiveness of enforcing the Dummy Consistency axiom in the experiments .

3.	Limited Analysis of Generalization Across Tasks:
The experiments on text tasks only compare entropy on BERT-Large (IMDb) and claim sparse attribution improvements. However, similar analyses are not conducted for image tasks. Moreover, the comparison of attribution methods through other metrics is absent on text tasks. A broader analysis is needed to demonstrate the generalizability and effectiveness of the proposed method across various tasks, including image and text domains .

4.	Formatting Issues:
The symbol τ is used in different contexts throughout the manuscript: once as a gradient threshold and once as a dummy threshold. To avoid confusion, I recommend differentiating the symbols, for example, by using τ_g for the gradient threshold and τ_dum for the dummy threshold.

**Questions:**

See weaknesses

---

> ### Author Response · Authors · 2025-11-26
> **Author Response to Reviewer N75c**
>
> Thank you for the detailed review and constructive feedback on our work. We especially appreciate your valuable comments and suggestions for further improving the work. Please find our point-by-point response below, indexed as [C1] through [C4].
>
> **[C1] “Questionable Table 1 Results (DiffID)”**
>
> We thank the reviewer for pointing out the issue with the standard deviation values of the DiffID scores in Table 1 and Table 5 in the appendix. We have recalculated the DiffID scores and revised the DiffID columns in Table 1 of the manuscript and Table 5 in Appendix C.6 as follows:
>
> **Table 1:** Updated DiffID Scores (ImageNet-1K)
>
> | method_name | VGG16 | IncV3 | RN18 |
> | --- | --- | --- | --- |
> | IG | 0.012 ± 0.01 | 0.061 ± 0.21 | 0.025 ± 0.08 |
> | EG | 0.014 ± 0.02 | 0.066 ± 0.38 | 0.025 ± 0.10 |
> | Saliency | 0.010 ± 0.01 | 0.065 ± 0.36 | 0.023 ± 0.08 |
> | IxG | 0.011 ± 0.01 | 0.068 ± 0.35 | 0.026 ± 0.10 |
> | FG | 0.041 ± 0.04 | 0.083 ± 0.31 | 0.059 ± 0.24 |
> | DGA | 0.046 ± 0.05 | 0.090 ± 0.38 | 0.061 ± 0.20 |
> | AGI | 0.019 ± 0.02 | 0.059 ± 0.19 | 0.028 ± 0.09 |
> | RFA | 0.038 ± 0.05 | 0.092 ± 0.39 | 0.049 ± 0.16 |
> | LPI | 0.013 ± 0.01 | 0.058 ± 0.25 | 0.025 ± 0.09 |
> | DCAPM | 0.036 ± 0.06 | 0.098 ± 0.64 | 0.059 ± 0.25 |
>
> **Table 5:** Updated DiffID Scores (Appendix C.6)
>
> | method_name | flowers | oxfordpet |
> | --- | --- | --- |
> | IG | 0.014 ± 0.012 | 0.004 ± 0.003 |
> | EG | 0.022 ± 0.019 | 0.005 ± 0.004 |
> | Saliency | 0.011 ± 0.009 | 0.003 ± 0.002 |
> | IxG | 0.018 ± 0.019 | 0.004 ± 0.003 |
> | FG | 0.061 ± 0.046 | 0.027 ± 0.022 |
> | DGA | 0.061 ± 0.044 | 0.028 ± 0.024 |
> | AGI | 0.023 ± 0.025 | 0.006 ± 0.005 |
> | RFA | 0.044 ± 0.039 | 0.023 ± 0.021 |
> | DCAPM | 0.067 ± 0.044 | 0.015 ± 0.022 |
>
> **[C2] “Lack of Empirical Statistical Analysis for Dummy Consistency”**
>
> We would like to respectfully clarify that we have already conducted a quantitative statistical analysis to evaluate the Dummy Consistency property. Specifically, we proposed a metric in Section 4 (Eq. 11) designed to measure the violation of Dummy Consistency. Using this metric, we reported the statistical results in Table 1: DCAPM (Ours) demonstrates significantly lower violation scores compared to the baselines. This empirical evidence supports our theoretical claim that DCAPM effectively removes spurious interactions and strictly adheres to the Dummy Consistency axiom during path optimization.
>
> **[C3] “Limited Analysis of Generalization (Sparsity Analysis)”**
>
> To demonstrate the generalization of our method's effectiveness, we included an entropy analysis across the image domain in Appendix C.5. Figure 9 illustrates that DCAPM achieves a significantly lower mean entropy (12.46) compared to baselines (14.71–15.52). This confirms that our method yields consistently sparse explanations across different samples, effectively filtering out noise.
>
> **[C4] “Formatting Issues the symbol $\tau$”**
>
> We agree that using the symbol $\tau$ for both the gradient threshold and the dummy threshold could lead to confusion. Following your suggestion, we have revised the manuscript to use distinct symbols: $\tau_g$ for the gradient threshold; $\tau_{dum}$ for the dummy threshold.

---

### Official Review · Reviewer_8xAS · 2025-10-31

**Soundness:** 1
**Presentation:** 3
**Contribution:** 1
**Rating:** 2
**Confidence:** 4

**Summary:**

The paper introduces a method of post-hoc, gradient based attribution in the path-method vein. The primary motivation of the paper is to craft an attribution method that does not attribute non-zero value to "dummy features", or features that are independent of label truth. To this end, the paper purports to illustrate how existing methods do report non-zero attribuiton to dummy features, motivates why attribution methods should not have this issue, and then outlines a method that avoids the issue: dummy-consistent attribution path method (DCAPM). Experimental results comparing attribuiton methods are provided, showing desirable performance for DCAPM.

**Strengths:**

- Descent (rigorous and clear) methematical presentation
- Positive experimental results
- descently clear writing most of the time.

**Weaknesses:**

- Critical issues in the treatment of the dummy consistency axiom
- The method seeks to address dummy consistency, but does not engage in an axioms-oriented development of the method:
  - the priority of the paper is not to find methods that satisfy a set of desirable axioms
  - the paper provides no characterization results (only my method satisfies this set of axioms), which is the strongest characteristic of so-called axiomatic attributions methods such as SHAP and IG.
- Some language is loose
  - focus on Shapley Value in beginning is misleading, as it is non-central to the paer compared to IG.
  - terms common in the attribution-discussion, like "shattered gradeints", are used to explain phenomena, but not treated rigorously.

**Questions:**

- Critial issues with the treatment of dummy:
  1) The paper confuses the following two concepts: 1) a feature is irrelevant to the true label of an image, and 2) a feature has no effect on the model's categorization of an input.
  2) the paper claims that an attribution method violates dummy consistency if there is a category-2 feature that the attibution method assigns non-zero attribution to.
  3) due to the above confusion, the paper claims that an attribution method does not satisfy dummy consistency becase a category-1 feature does not have zero attribution.

We see this in lines 201-203, where ithe paper claims that some attribution methods eroniously agggregate/report noisy gradients of irrelevant features. However, those features are presumably category-1 features; they certainly are not category-2 features.

The purpose of an attribution method is to explain the model, not to report what we believe the model should be doing. We do not entirely know what the model is doing, it could be using what we believe are spurious features. Thus, if the attribution method reports that a category-1 feature is relevant, it is difficult for us to say we know the model is not significantly using it. Even if we design a feature to be irrelevant (feature is noise), when we train a model it will try to predict using the irrelevant feature; intentionally irrelevant features usually do cause an effect in models. Thus, when an attributon reports that these features are contributers, how do we know the attribution method is reporting erroniously?

- The author at times tries to get around the above issue buy using a threshold assumption - if the gradients of a feature is significantly and consistently small, then we can assume that the feature's effect on the model is adding small-magnitude irrelevant noise to the output. I would agree that if a feature is small-magnitude irrelevant noise then the gradients are likely to be small and irregular (still needs some justification). However, I do not agree that if gradients are small and irredular, then the feature is irrelevant. What if a consistently small-gradient feature is important to a model's prediction as a small and perhaps noisy contributer? In this case, DCAPM would attribute zero to contributing features, mistaking them for irrelevant features.

- To be clear, the method seems to have some merit in reducing spurious correlations and shattered gradients (it is theoretically suggestive and via experiemntal results), but the interaction of the paper with the axiomatic attributions feamework and dummy consistency needs more careful consideration and proper hedging, perhaps in cooperation with a different framing and motivation that emphasizes the axioms less.

- I found figure 1 more confusing than illuminating. What is the color coding of the first/second columns? What are are the regions detailing? Why exactly are they being warped? Is the only difference between row 1 and 2 IG vs FG? If so, why are they different when no attributions are plotted, and what is the color difference between them? What are the black, white, and green dots? When x2 is intoduces, was the model retrained?

- Equation 6 uses an i subscript, but does not depend on i. Perhaps I am misunderstanding something?

---

> ### Author Response · Authors · 2025-11-26
> **Author Response to Reviewer 8xAS (1/2)**
>
> Thank you for the detailed review and constructive feedback on our work. We especially appreciate the reviewer pointing out the clarity issue regarding our motivation and claims. Please find our point-by-point response below, indexed as [C1] through [C6].
>
> **[C1] “Some language is loose”**
>
> We fully agree with the reviewer that emphasizing the Shapley value at the beginning was misleading, as the core of our work is much more directly related to IG and the Aumann–Shapley (AS) path framework. Our original intention was not to claim that our method is *grounded* in the Shapley value itself, but rather to highlight the following point:
>
> > Even though IG and other AS-based methods satisfy axioms such as Dummy Consistency in theory, they often fail to satisfy these axioms in practice when applied to rectified networks, primarily due to the shattered gradient phenomenon.
> >
>
> To avoid the confusion pointed out by the reviewer, we have revised the abstract, introduction and Sec. 2 to (1) remove unnecessary emphasis on the Shapley value, (2) clarify and tone down  that our contribution is not offering a new axiomatic characterization, and (3) refocus the discussion on the *practical* violation of dummy-related axioms in standard path methods.
>
> **[C2] “Clarification on shattered gradients”**
>
> We acknowledge the need for rigor in defining the term "shattered gradients." We have clarified the phenomenon more precisely in Section 2.2 as follows:
>
> > The shattered gradients problem is the phenomenon observed in deep networks where the spatial correlation between gradients at nearby data points decays exponentially with network depth (Balduzzi et al., 2017). As a result, the gradient signal loses its local structure and increasingly resembles uncorrelated white noise. The shattering of gradients is a direct consequence of the composition of rectifier non-linearities across many layers, due to factors such as piecewise constant gradients and the exponential growth of activation boundaries.
> >
>
> **[C3] “Confusion on Dummy Definitions (Category 1 vs. 2)”**
>
> We appreciate the reviewer highlighting this ambiguity and fully agree with the distinction. We fully agree that Category 1 (features irrelevant to the true label of an image) and Category 2 (features that have no functional effect on the model output) must be clearly separated. Our method is explicitly designed to identify and suppress Category 2 features. The objective is to correct practical failure cases where gradients become non-zero due to noise (shattered gradients), even though the model’s marginal dependence on that feature is negligible. Therefore, while our method often removes background noise (Category 1) as a byproduct, it does so only because the model is not functionally using those features.
>
> To further clarify this point, we added an additional experiment in Appendix C.4 demonstrating that our method correctly preserves spurious correlations (Category 1 features that the model actually uses). This confirms that DCAPM faithfully reflects the model's behavior rather than imposing a prior to ignore background. We have also revised phrases in the Introduction and Results sections to avoid implying that "semantic irrelevance" is our criterion for removal.
>
> **[C4] “Small and irregular gradients ≠ irrelevance”**
>
> We appreciate this important point. We agree that gradient magnitude alone is not a reliable indicator of feature relevance. DCAPM, however, does not classify features using a simple threshold. Instead, it uses an optimization process to evaluate whether a feature is functionally necessary. The loss function in Eq. 3 contains a prediction preservation term that directly measures the effect of masking. When a feature plays even a small but genuine role in the model output, masking it results in a measurable increase in prediction error. This effect dominates the sparsity term, compelling the optimization to keep the feature active. Thus, DCAPM does not remove features simply because their gradients are small. It removes only those features whose masking produces no meaningful change in the model output—indicating true functional irrelevance.
>
> **[C5] “Figure 1 is more confusing than illuminating”**
>
> We added specific legends for 'loss' and 'logits' to the colorbar in Column (a). We replaced the ambiguous colored dots with numbered points (1–5) to clearly indicate the inputs for the single-axis-shift scenario. Additionally, we applied a divergent 'blue-white-red' colormap in Column (d) to intuitively visualize negative, zero, and positive values, and explicitly marked the colors corresponding to the minimum and maximum values in the legend.

---

> ### Author Response · Authors · 2025-11-26
> **Author Response to Reviewer 8xAS (2/2)**
>
> **[C6] “Equation 6 uses an i subscript, but does not depend on i.”**
>
> The index $i$ in $\mathcal{I}_{i,j}$ (which denotes the interaction between features $i$ and $j$) is implicitly tied to the update step.  eature $i$ represents the active, non-dummy feature in the set $S^{(t-1)}$ that drives the current path update. The conceptual intention was to denote that the variation observed in the dummy feature $j$ is *caused* by the introduction of feature $i$. We have clarified in the text that while the RHS is computed via the gradient difference at step $t$, this step corresponds specifically to the contribution of feature $i$.

---

### Author Response · Authors · 2025-12-02
**Summary of Revisions and Response to Reviewers**

Dear Area Chairs,

We sincerely appreciate the constructive feedback provided by the reviewers. We recognize that the initial concerns primarily stemmed from the clarity of our motivation regarding "Dummy Consistency" and the need for broader empirical validation.

In response, we have engaged in extensive discussions and implemented significant revisions to address these points. Below, we summarize the key strengths recognized by the reviewers and the major improvements made to the manuscript.

### **1. Key Strengths Acknowledged by Reviewers**

- **Important Problem Definition:** Reviewers highlighted that addressing "Dummy Consistency" is a crucial and relatively under-discussed topic in explainable AI (Addressing Reviewers RB8P and N75c).
- **Rigorous Mathematical Presentation:** The paper offers a rigorous mathematical formulation (8xAS) and an intuitive mask-based optimization strategy for finding clean baselines (N75c).
- **Novelty:** The proposed DCAPM is recognized for its novel approach to identifying relevant features (to prediction) by masking, which effectively lowers prediction error (FCW1).

### **2. Summary of Major Revisions & Clarifications**

To address the concerns raised, we have substantially strengthened the paper in the following dimensions:

**1) Clarification of Core Concepts** (Addressing Reviewers 8xAS and FCW1)

- **Definition of Dummy Features:** We clarified the distinction between 'semantically irrelevant features' and 'functionally irrelevant features' (referred to as Category 1 and 2 by Reviewer 8xAS). We emphasized that DCAPM is designed to suppress features that have no functional effect on the model but receive non-zero attribution due to shattered gradients in deep rectified networks (See Response C3 for 8xAS).
- **Visualization of Motivation and DCAPM’s Effect:** We updated the symbols and colormaps in the motivation figure (**Figure 1**) to resolve confusion(See Response C5 for 8xAS). Furthermore, we additionally visualized the effect of DCAPM on a 2D dataset in **Figure 7 (Appendix C.2)** and **Figure 13 (Appendix C.9)**, explicitly demonstrating its ability to nullify noise feature attribution and spurious attributions arising from interactions between dummy features (See Response C5 for 8xAS and C1 for FCW1).
- **Mechanism Justification:** We elaborated on why iterative path optimization is necessary over simple baselines (e.g., Input $\times$ Gradient). We clarified that static methods fail to capture how feature importance evolves and interacts, whereas DCAPM dynamically filters transient noise. We demonstrate that DCAPM outperforms Input $\times$ Gradient in the quantitative comparisons presented in **Table 1** of the main text and **Table 7 of Appendix C.7**. (See Response C2 for FCW1).

**2) Expanded Experimental Scope** (Addressing Reviewers RB8P & N75c)

- **Spurious Correlation Analysis (Waterbirds):** As requested, we added experiments on the Waterbirds dataset (**Appendix C.5**) to demonstrate DCAPM’s effectiveness in identifying spurious correlations, confirming it does not blindly remove background features if the model relies on them (See Response C6 for RB8P).
- **Modern Architectures:** We validated DCAPM on ConvNeXt-T (Liu et al., 2022), demonstrating its applicability to modern architectures beyond standard CNNs (**Appendix C.4)** (See Response C5 for RB8P).
- **Generalization Analysis:** We added an entropy analysis (**Appendix C.6**) to demonstrate the method's consistent sparsity and noise-filtering capabilities across different samples (See Response C3 for N75c).

**3) Data Correction and Physical Interpretation** (Addressing Reviewers N75c and RB8P)

- **Recalculated Metrics:** We corrected the calculation error regarding the standard deviation of DiffID scores in **Table 1** and **Table 6**, ensuring accurate statistical reporting (See Response C1 for N75c).
- **Physical Meaning of Path:** We revised the manuscript to explicitly describe the "path" and the physical meaning of the loss function terms (driving force for removal vs. resistance force for sparsity). We also revised the manuscript (Lines 97–98) to explicitly direct readers to the path visualization provided in **Figure 2 Row 2**, which vividly illustrates the concept (See Response C1 and C2 for RB8P) .

We believe these revisions comprehensively address the reviewers' concerns and significantly enhance the robustness and clarity of our contribution. We respectfully ask the AC to consider these improvements in the final decision.

Best regards,

Authors

---

### Meta-Review · Area_Chair_PfaA · 2026-01-05

**Summary:**

This paper introduces the Dummy Consistent Attribution Path Method (DCAPM), a novel gradient-based attribution technique that aims to eliminate the issue of assigning non-zero attribution to "dummy features". The paper emphasizes how current attribution methods, such as Integrated Gradients (IG), fail to satisfy the dummy consistency axiom in practice due to issues like shattered gradients. DCAPM seeks to address these issues by constructing a path that optimizes feature relevance along a trajectory from a baseline to the original input. Experimental results suggest that DCAPM outperforms traditional methods in terms of both attribution accuracy and interpretability.

**Reviewer Concerns:**

The reviewers raised several critical concerns, particularly regarding the handling of dummy consistency and the clarity of the paper's motivations. Reviewer 8xAS pointed out confusion regarding the distinction between category-1 and category-2 features and their relationship to small gradients. Reviewer N75c expressed concerns over the questionable DiffID results, particularly the large standard deviation, and the lack of sufficient statistical analysis to support the claims of enforcing dummy consistency. Reviewer FCW1 raised issues with the motivation behind the iterative optimization procedure and its complexity, suggesting that simpler approaches might suffice. Finally, Reviewer RB8P highlighted the unclear physical meaning of the "integration path," issues with the pairwise feature interactions, and the failure to address the importance of higher-order interactions, which are critical to the method’s applicability. Furthermore, there was dissatisfaction with the comparison to DGA and the lack of new experimental results for the ConvNeXt and Waterbird datasets.

**Reviewer Scores:**

Reviewer 8xAS raised a concern regarding the confusion between "a feature is irrelevant to the true label of an image" and "a feature has no effect on the model's categorization of an input." The authors tried to clarify this distinction, but the explanation of how small gradients for potentially relevant features are handled was insufficient. This ambiguity leaves the concern unresolved, and the reviewer is likely to keep their rating unchanged. And Reviewer FCW1 questioned the necessity of the iterative optimization procedure over simpler static methods, asking why the latter wouldn't suffice. The authors provided an explanation, but the rationale was not fully convincing. This lack of clarification leaves the concern unaddressed, and the reviewer would likely not change their rating.

---

### Decision · Program_Chairs · 2026-01-26

Reject